# Mechanically stimulated osteocytes maintain tumor dormancy in bone metastasis of non-small cell lung cancer by releasing small extracellular vesicles

Jing Xie[1,2†], Yafei Xu[1†], Xuhua Liu[3], Li Long[2], Ji Chen[2], Chunyan Huang[2], Yan Shao[3], Zhiqing Cai[2], Zhimin Zhang[2], Ruixin Zhou[2], Jiarong Leng[4], Xiaochun Bai[2,3]*, Qiancheng Song[2,4]*

[1]General Practice Centre, The Seventh Affiliated Hospital, Southern Medical University, Foshan, China; [2]Guangdong Provincial Key Laboratory of Bone and Joint Degeneration Diseases, Department of Cell Biology, School of Basic Medical Sciences, Southern Medical University, Guangzhou, China; [3]Academy of Orthopedics, Guangdong Province, Guangdong Provincial Key Laboratory of Bone and Joint Degeneration Diseases, The Third Affiliated Hospital of Southern Medical University, Guangzhou, China; [4]Department of Neurosurgery, Institute of Brain Diseases, Nanfang Hospital of Southern Medical University, Guangzhou, China

*For correspondence:
baixc15@smu.edu.cn (XB);
songqc@smu.edu.cn (QS)

[†]These authors contributed equally to this work

**Abstract** Although preclinical and clinical studies have shown that exercise can inhibit bone metastasis progression, the mechanism remains poorly understood. Here, we found that non-small cell lung cancer (NSCLC) cells adjacent to bone tissue had a much lower proliferative capacity than the surrounding tumor cells in patients and mice. Subsequently, it was demonstrated that osteocytes, sensing mechanical stimulation generated by exercise, inhibit NSCLC cell proliferation and sustain the dormancy thereof by releasing small extracellular vesicles with tumor suppressor micro-RNAs, such as miR-99b-3p. Furthermore, we evaluated the effects of mechanical loading and treadmill exercise on the bone metastasis progression of NSCLC in mice. As expected, mechanical loading of the tibia inhibited the bone metastasis progression of NSCLC. Notably, bone metastasis progression of NSCLC was inhibited by moderate exercise, and combinations with zoledronic acid had additive effects. Moreover, exercise preconditioning effectively suppressed bone metastasis progression. This study significantly advances the understanding of the mechanism underlying exercise-afforded protection against bone metastasis progression.

## eLife assessment

This is an **important** study, that adds to the field a new understanding of exercise or mechanical loading, microRNAs, and secreted extracellular vessicles in the field of lung cancer (NSCLC), which may have relevance to other osteolytic cancers. The strength of the evidence was mixed: whereas in vitro microRNA experiments were **convincing**, other elements were **incomplete** (e.g., proving the roles of osteocytes, as opposed to other mechanosensitive cells, in vivo). This work would be of broad interest to those investigating osteolytic cancers, and the role of exercise in bone cancer, preclinically.

## Introduction

Lung cancer is the second most common cancer type and the leading global cause of cancer mortality (18.0% of the total cancer deaths) (*Sung et al., 2021*). Approximately 85% of lung cancer patients have a group of histological subtypes collectively termed non-small cell lung cancer (NSCLC) (*Herbst et al., 2018*). Despite important advancements in NSCLC treatment that have been achieved over the past two decades, prognosis remains poor owing to the presence of locally advanced or widely metastatic tumors in most patients at the time of diagnosis (*Herbst et al., 2018*). Approximately 20–30% of patients with NSCLC present with bone metastases at diagnosis, and 35–60% will develop them during their disease course (*Cetin et al., 2014*; *Riihimäki et al., 2014*). Bone metastases are frequent events associated with advanced-stage malignancies and often cause skeletal-related events (SREs) such as severe bone pain, pathological fractures, spinal cord compression, and hypercalcemia, which reduce the quality of life and predict a poor prognosis (*Cetin et al., 2014*; *Sowder and Johnson, 2019*; *Zhang et al., 2019*). In general, bone metastases are incurable; goals of management include maximizing symptom control, preserving function, minimizing SREs, and enhancing local tumor control (*Herbst et al., 2018*; *Knapp et al., 2022*). A multidisciplinary treatment approach is required, including pain management, osteoclast inhibitors and other bone-specific therapy, systemic anti-cancer therapy, surgery, radiation, and interventional techniques. Although there have been many advances in treatment, the median survival from diagnosis of bone metastases has been reported to be very short in patients with lung cancer (*Herbst et al., 2018*; *Knapp et al., 2022*). Therefore, it is of considerable clinical significance to elucidate the mechanism of bone metastasis and to find new drugs and combination therapies for bone metastasis prevention and treatment in NSCLC.

Many preclinical and clinical studies have shown that exercise can reduce the incidence of various cancers, inhibit cancer progression, reduce symptom burden, mitigate side effects of cancer treatment, and improve survival and the quality of life in patients with cancer (*Bade et al., 2015*; *Ligibel et al., 2022*; *Pedersen et al., 2016*; *Schmitz et al., 2019*). In contrast, sedentary behavior and lack of physical activity is positively associated with an increased risk for cancer and mortality (*Cao et al., 2022*; *Kerr et al., 2017*). Exercise is safe perioperatively, and multiple international organizations have published exercise recommendations for patients with cancer (*Bade et al., 2015*; *Ligibel et al., 2022*; *Stout et al., 2021*). The evidence to-date suggests a benefit of exercise in all stages of lung cancer and lung cancer survivors (*Bade et al., 2015*). Exercise has the potential to improve physical function and the quality of life in individuals with bone metastases and appears safe and feasible when it includes an element of supervised exercise instruction (*Ligibel et al., 2022*; *Weller et al., 2021*). The optimal exercise prescription for patients with bone metastases is currently undefined (*Cavalheri and Granger, 2020*; *Stout et al., 2021*). Each patient with cancer may require a more personalized exercise prescription. Therefore, a clearer understanding of the mechanisms by which exercise modulates tumors is required.

Exercise induces systemic changes involving every system over the entire body, including circulating substances, the immune system, and metabolism (*Ruiz-Casado et al., 2017*; *Wang and Zhou, 2021*; *Zhu et al., 2022*). Throughout life, the skeleton undergoes continuous remodeling, which is a balance between the resorption of mineralized bone by bone-resorbing cells (osteoclasts) and new bone formation by bone-forming cells (osteoblasts) (*Wang et al., 2022*). Osteocytes, the most abundant cell type in bone, are terminally differentiated osteoblasts that have become buried in their own matrix. During this process, their morphology changes to stellate, facilitating the formation of an extensive interconnected network throughout the bone matrix termed the lacunar-canalicular network (LCN) (*Prideaux et al., 2016*; *Robling and Bonewald, 2020*; *Schaffler et al., 2014*). Through the LCN, osteocytes sense mechanical stimulus and translate them into biochemical signals, and appropriately control bone homeostasis by regulating the function of both osteoclasts and osteoblasts. Therefore, the osteocyte is now recognized as a major orchestrator of the bone remodeling process. However, the understanding of the relationship between osteocytes and bone metastasis remains limited.

Bone metastasis is a multistage process. Long before the development of clinically detectable metastases, circulating tumor cells settle and enter a dormant state in normal vascular and endosteal niches present in the bone marrow, which provide immediate attachment and shelter, and the cells only become active years later as they proliferate and alter the functions of osteoclasts and osteoblasts, promoting skeletal destruction (*Clézardin et al., 2021*). The endosteal niche, including osteoblasts and osteoclasts, has been reported to provide tumor cells with an environment supporting

their survival and outgrowth, however, the role of osteocytes has been rarely addressed (*Zhang et al., 2019*). Tumor cell dormancy is defined as an arrest in the cell cycle (*Blasco et al., 2022*; *Phan and Croucher, 2020*). The details of niches and signals that support tumor cell dormancy in bone metastasis are largely unknown. In the current study, we discuss the effect of exercise on bone metastasis of NSCLC, and investigated the potential effects and mechanisms of the osteocyte on the dormancy of bone metastasis NSCLC cells.

## Results

### Osteocytes inhibited NSCLC cell proliferation

To study the spatial distribution of NSCLC cell proliferation in bone metastasis, bone metastasis tissues of six NSCLC patients were obtained. Bone destruction was found in the presence of bone metastasis in NSCLC (*Figure 1—figure supplement 1a*). Surprisingly, immunohistochemical staining showed that Ki-67 was much lower in tumor cells adjacent to bone tissue than in the surrounding tumor cells (*Figure 1a*). An intraosseous model was used to investigate the biological progression of NSCLC cells in a bone microenvironment via direct tibial implantation of A549 in nude mice. The results of biophotonic imaging, micro-computed tomography (micro-CT), and hematoxylin and eosin (H&E) staining indicated that the model was successfully established 21 days after modeling (*Figure 1—figure supplement 1b–d*). Consistent with the result in the clinical samples, Ki-67 expression was much lower in tumor cells adjacent to bone tissue than in the surrounding tumor cells (*Figure 1b*). The above results indicated that NSCLC cells adjacent to bone tissue have a much lower proliferative capacity than the surrounding tumor cells. Therefore, we hypothesized that bone tissues release some substance that inhibits tumor growth.

To determine which type of cell inhibits proliferation after NSCLC cells have invaded bone tissue, A549 was co-cultured with MLO-Y4, MC3T3-E1, and NCM460 (as a negative control) cells for 48 hr, respectively (*Figure 1c*). We found that the A549 migration ability was improved after co-culture with MLO-Y4, with the promotional effect being much significantly stronger on co-culture with MC3T3-E1 (*Figure 1—figure supplement 1e*). More importantly, NSCLC cell proliferation was inhibited after co-culture with MLO-Y4, whereas no inhibitory effect was found after co-culture with MC3T3-E1 (*Figure 1d and e* and *Figure 1—figure supplement 1f*), indicating that it is osteocytes that inhibit NSCLC cell proliferation. Moreover, MLO-A5, another osteocyte-like cell, also inhibited the proliferation of NSCLC cells (*Figure 1—figure supplement 1g*). Surprisingly, MC3T3-E1 proliferation was effectively enhanced on co-culture with MLO-Y4 (*Figure 1f*), which indicated that the effect of osteocytes on NSCLC cell and osteoblast proliferation were different. Taken together, these data suggested that it is osteocytes that inhibit NSCLC cell proliferation in the endosteal niche.

### Osteocyte sEVs inhibited NSCLC cell proliferation

To investigate the mechanism by which osteocytes inhibit NSCLC cell proliferation, osteocyte-conditioned medium (CM) was separated into soluble factor (SF), large extracellular vesicles (lEVs), and small extracellular vesicles (sEVs) fractions by serial ultracentrifugation and ultrafiltration as performed previously with some modifications (*Fafián-Labora et al., 2020*; *Figure 2a*). The purified sEVs were verified by transmission electron microscopy (TEM), NanoSight analysis, and immunoblotting for protein markers (CD9, Alix, and TSG101), according to the proposal of the International Society of Extracellular Vesicles (*Figure 2—figure supplement 1a–c*). sEVs were labeled with PKH67 and were observed to be internalized by NSCLC cells (*Figure 2—figure supplement 1d*). Subsequently, we treated the NSCLC cells with fractions derived from osteocyte CM, with EV-depleted media as a negative control (fetal bovine serum [FBS] 10%). Interestingly, sEVs from osteocyte CM had a much significantly greater potential to inhibit NSCLC cell proliferation than the other groups (*Figure 2—figure supplement 1e and f*), whereas sEVs from osteoblast CM had no inhibitory effect on NSCLC cell proliferation (*Figure 2b and c*).

To confirm the role of sEVs in osteocyte inhibition of NSCLC cell proliferation, GW4869 was used to inhibit sEVs secretion. The significant reduction of sEVs released from MLO-Y4 on GW4869 treatment was confirmed by NanoSight analysis and western blotting (*Figure 2d and e*). More importantly, it was found that GW4869 effectively eliminated the inhibitory effect of MLO-Y4 on NSCLC cell proliferation (*Figure 2f and g*), which suggested that sEVs release was required for osteocytes to inhibit

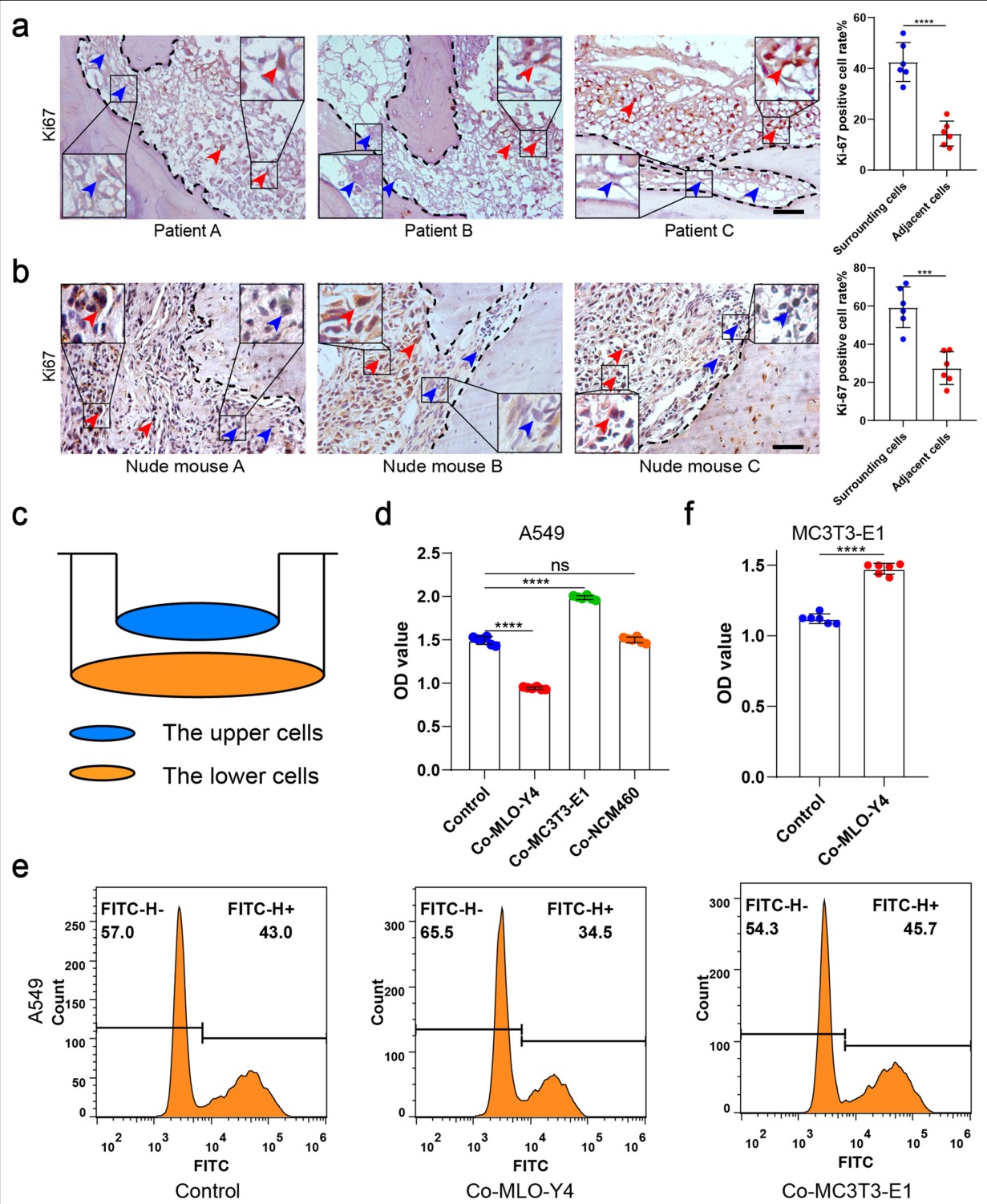

**Figure 1.** Osteocytes inhibited the proliferation of non-small cell lung cancer (NSCLC) cells. (**a**) Representative immunohistochemical staining and quantification of Ki-67 in bone metastasis site of patients with NSCLC. The dashed line indicates the boundary between bone and tumor. Scale bar: 50 μm. Blue arrows, low expression or negative of Ki-67. Red arrows, high expression of Ki-67. Tumor cells ≤50 μm from bone tissue were defined as adjacent cells, and those >50 μm from bone tissue were defined as surrounding cells. n=6; Student's two-sided unpaired t test. (**b**) An intraosseous model of bone metastasis was used via direct implantation of NSCLC cells A549 into tibia of nude mice. Three weeks after implantation, mice were sacrificed. Representative immunohistochemical staining and quantification of Ki-67 in tibia of tumor-bearing nude mice. The dashed line indicates the boundary between bone and tumor. Scale bar: 50 μm. Blue arrows, low expression or negative of Ki-67. Red arrows, high expression of Ki-67. Tumor

*Figure 1 continued on next page*

*Figure 1 continued*

cells ≤50 μm from bone tissue were defined as adjacent cells, and those >50 μm from bone tissue were defined as surrounding cells. n=6; Student's two-sided unpaired t test. (**c**) Schematic diagram of co-culture. (**d**) Cell Counting Kit-8 (CCK-8) assays were performed to evaluate the effects of MLO-Y4 cells, MC3T3-E1 or NCM460 cells on the proliferation of A549 cells in the co-culture system. n=6; one-way analysis of variance with Turkey's multiple comparisons test. (**e**) EdU flow cytometry were performed to evaluate the effect of MLO-Y4 cells on the proliferation of A549 cells in the co-culture system. (**f**) CCK-8 assays were performed to evaluate the effect of MLO-Y4 cells on the proliferation of MC3T3-E1 cells in the co-culture system. n=6; Student's two-sided unpaired t test. Error bars represent Mean ± SD. No significance (ns) p>0.05, *p<0.05, **p<0.01, ***p<0.001, ****p<0.0001.

The online version of this article includes the following source data and figure supplement(s) for figure 1:

**Source data 1.** Original table sources for quantification of *Figure 1* plots.

**Figure supplement 1.** Histological identification of patients and mice with bone metastases and effects of MLO-Y4 and MC3T3-E1 on proliferation and migration of non-small cell lung cancer (NSCLC) cells.

**Figure supplement 1—source data 1.** Original table sources for quantification of *Figure 1—figure supplement 1* plots.

NSCLC cell proliferation. Due to the complexity of the bone marrow microenvironment, we examined the effect of bone marrow-derived sEVs on NSCLC cells proliferation by Cell Counting Kit-8 (CCK-8) assay. The results showed that bone marrow-derived sEVs could promote the proliferation of NSCLC cells (*Figure 2—figure supplement 1g, h*). Taken together, osteocytes inhibit NSCLC cell proliferation by releasing sEVs.

## Osteocyte sEVs miR-99b-3p inhibited NSCLC cell proliferation by directly targeting murine double minute2

Subsequently, we performed micro-RNA (miRNA) sequencing to investigate how osteocyte-derived sEVs inhibit NSCLC cell proliferation. The miRNAs, which can be abundantly encapsulated in sEVs, are key regulators of cancer cell proliferation. To identify the specific miRNAs involved, we compared the miRNA profiles of MLO-Y4-sEVs and MC3T3-E1-sEVs, the results of which are shown as heat-maps (*Figure 3—figure supplement 1a*). Four of the maximally upregulated miRNAs and two of the downregulated miRNAs were validated by real-time quantitative polymerase chain reaction (qRT-PCR) (*Figure 3a*).

These four upregulated miRNA sequences are identically conserved in the human and mouse. To determine the effects of these miRNAs on NSCLC cell proliferation, A549 and NCI-H23 cells were transfected with miRNA mimics or negative controls. We found that both miR-99b-3p and miR-193a-5p significantly inhibited NSCLC cell proliferation 48 hr after transfection (*Figure 3b*), with miR-99b-3p having the most significant inhibitory effect. Inhibitors of miR-99b-3p, by contrast, promoted NSCLC cell proliferation (*Figure 3c and d*). These data indicated that sEVs released by osteocytes carried tumor suppressor miRNAs, such as miR-99b-3p.

Notably, in situ hybridization analysis demonstrated high miR-99b-3p expression in the cortical bone of patients with bone metastases of NSCLC (*Figure 3—figure supplement 1b*). To identify the targets of miR-99b-3p, first, we applied PicTar and TargetScan to search for putative targets, and >100 mRNAs were predicted to be regulated by miR-99b-3p. Among these candidates, six genes are involved in cell cycle or proliferation regulation. To determine whether miR-99b-3p targets these genes directly, we cloned the 3'UTRs of the putative target into a dual luciferase assay system. Murine double minute 2 (MDM2), as a negative regulator of p53, has been shown to exert oncogenic activity (*Karni-Schmidt et al., 2016*). A reporter assay using A549 cells revealed that miR-99b-3p significantly repressed MDM2 3'UTRs. Mutations of the putative miRNA sites in MDM2 3'UTRs abrogated the responsiveness to miRNA (*Figure 3e*). Furthermore, transfection of miR-99b-3p mimicked reduced endogenous MDM2 protein levels in A549 cells. By contrast, inhibition of miR-99b-3p significantly increased the MDM2 protein level (*Figure 3f*).

Taken together, these results suggested that miR-99b-3p derived from osteocyte sEVs directly downregulates MDM2 expression through its 3'UTR, by which osteocytes inhibited NSCLC cell proliferation.

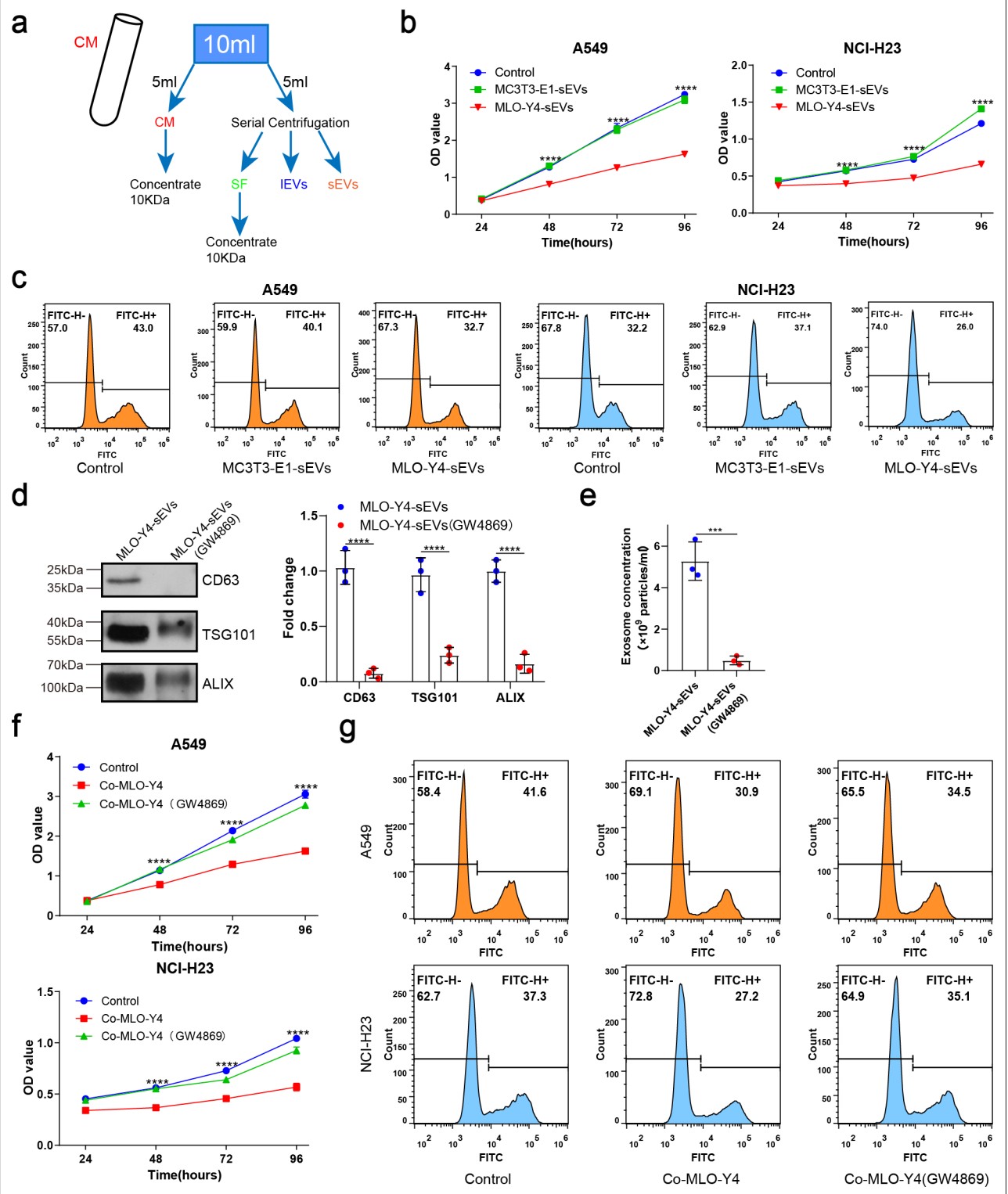

**Figure 2.** Osteocyte small extracellular vesicles (sEVs) inhibited the proliferation of non-small cell lung cancer (NSCLC) cells. (**a**) Schematic diagram of serial centrifugation and concentration of culture medium. Conditioned medium (CM) was separated into soluble factor (SF), large extracellular vesicles (lEVs), and sEVs fractions, by serial ultracentrifugation and ultrafiltration. (**b**) Cell Counting Kit-8 (CCK-8) assays were performed to evaluate the effect of sEVs on the proliferation of NSCLC cells. n=4, two-way analysis of variance with Sidak's multiple comparisons test. (**c**) EdU flow cytometry were performed to evaluate the effect of sEVs on the proliferation of NSCLC cells. (**d**) Western blot analysis of the typical sEVs markers CD63, TSG101, and Alix. n=3; Student's two-sided unpaired t test. (**e**) The concentration of the sEVs were detected by NanoSight analysis. n=3; Student's two-sided

*Figure 2 continued on next page*

*Figure 2 continued*

unpaired t test. (**f**) CCK-8 assays were performed to evaluate the rescue effect of GW4869 on the proliferation of NSCLC cells co-cultured with MLO-Y4 cells. n=4, two-way analysis of variance with Sidak's multiple comparisons test. (**g**) EdU flow cytometry were performed to evaluate the rescue effect of GW4869 on the proliferation of NSCLC cells co-cultured with MLO-Y4 cells. Error bars represent Mean ± SD. No significance (ns) p>0.05, *p<0.05, **p<0.01, ***p<0.001, ****p<0.0001.

The online version of this article includes the following source data and figure supplement(s) for figure 2:

**Source data 1.** Original table sources for quantification of *Figure 2* plots.

**Source data 2.** Original western blot images for *Figure 2d*.

**Source data 3.** Original western blot images for *Figure 2d* with highlighted bands and sample labels.

**Figure supplement 1.** Identification of MLO-Y4-sEVs and MC3T3-E1-sEVs and their effects on the proliferation ability of non-small cell lung cancer (NSCLC) cells.

**Figure supplement 1—source data 1.** Original table sources for quantification of *Figure 2—figure supplement 1* plots.

**Figure supplement 1—source data 2.** Original western blot images for *Figure 2—figure supplement 1c*.

**Figure supplement 1—source data 3.** Original western blot images for *Figure 2—figure supplement 1c* with highlighted bands and sample labels.

## Mechanical stimulation increased osteocyte sEVs release and enhanced the inhibitory effect of osteocytes on NSCLC cell proliferation

Osteocytes are thought to be the principal mechanosensors of bone by translating mechanical stimuli into molecular signals and appropriately controlling the downstream remodeling balance. Exercise has the potential to improve physical function and the quality of life in individuals with bone metastases. To determine whether osteocytes inhibit tumor proliferation by receiving mechanical stimulation, MLO-Y4 cells were subjected to sinusoidal stretching at 12% strain at a 1.25 Hz frequency for 12 hr, and sEVs were extracted from the CM. Interestingly, sEVs released from mechanically stimulated MLO-Y4 cells significantly inhibited NSCLC cell proliferation compared to static osteocyte sEVs (*Figure 4a and b* and *Figure 4—figure supplement 1a, b*). Moreover, the mechanically stimulated osteocyte sEVs had the most significant effect on the proliferation of NSCLC cells compared with SF and lEV components (*Figure 4—figure supplement 1c and d*). Notably, the concentration of sEVs of mechanically stimulated osteocytes was much significantly higher than that of static osteocyte sEVs (*Figure 4c and d*). Moreover, the miR-99b-3p content of osteocyte sEVs increased significantly after stretching (*Figure 4e*).

To identify the effect of mechanical stimulation on bone metastasis of lung cancer in vivo, mechanical loading (1 N at 2 Hz for 10 min per day, 5 days per week for 4 weeks) was applied to the tibiae with bone metastasis induced by direct tibial implantation of GFP-LLC in mice. The biophotonic imaging showed that tumor growth was inhibited by mechanical loading (*Figure 4f*). Consistent with this, Ki-67 expression in NSCLC cells in the mechanical loading group was much lower than that of the control group (*Figure 4g*). Dentin matrix protein 1 (DMP1), specifically expressed in osteocytes, was used as a specific molecular marker of osteocytes in this study. Meanwhile, CCND3 and GFP were selected for co-localization staining to confirm the above phenomena, the results showed that mechanical loading could effectively inhibit tumor growth (*Figure 4—figure supplement 1e*). The micro-computed tomography (micro-CT) analysis of the proximal tibia showed that the mechanical loading effectively protected bone integrity and delayed its destruction in the presence of bone metastasis in NSCLC, specifically reflected in the bone volume/total volume (BV/TV), trabecular thickness (Tb. Th), and cortical thickness (Ct.Th) (*Figure 4h* and *Figure 4—figure supplement 1f*). Furthermore, we found that AntagomiR-99b-3p could effectively rescue the inhibitory effect on NSCLC cell proliferation, indicating that miR-99b-3p plays an important role in the mechanically inhibited proliferation of NSCLC cells (*Figure 4i–k* and *Figure 4—figure supplement 1g and h*). Taken together, these results suggested that mechanical stimulation enhanced the inhibitory effect of osteocytes on bone metastasis progression by increasing the release of sEVs containing proliferation-suppressive miRNA.

## Moderate exercise combined with zoledronic acid effectively suppressed bone metastasis progression of NSCLC

Exercise is one of the ways in which the body senses mechanical stress, so we evaluated the effect of exercise in a mouse bone metastasis model. An intraosseous mouse model was used to investigate

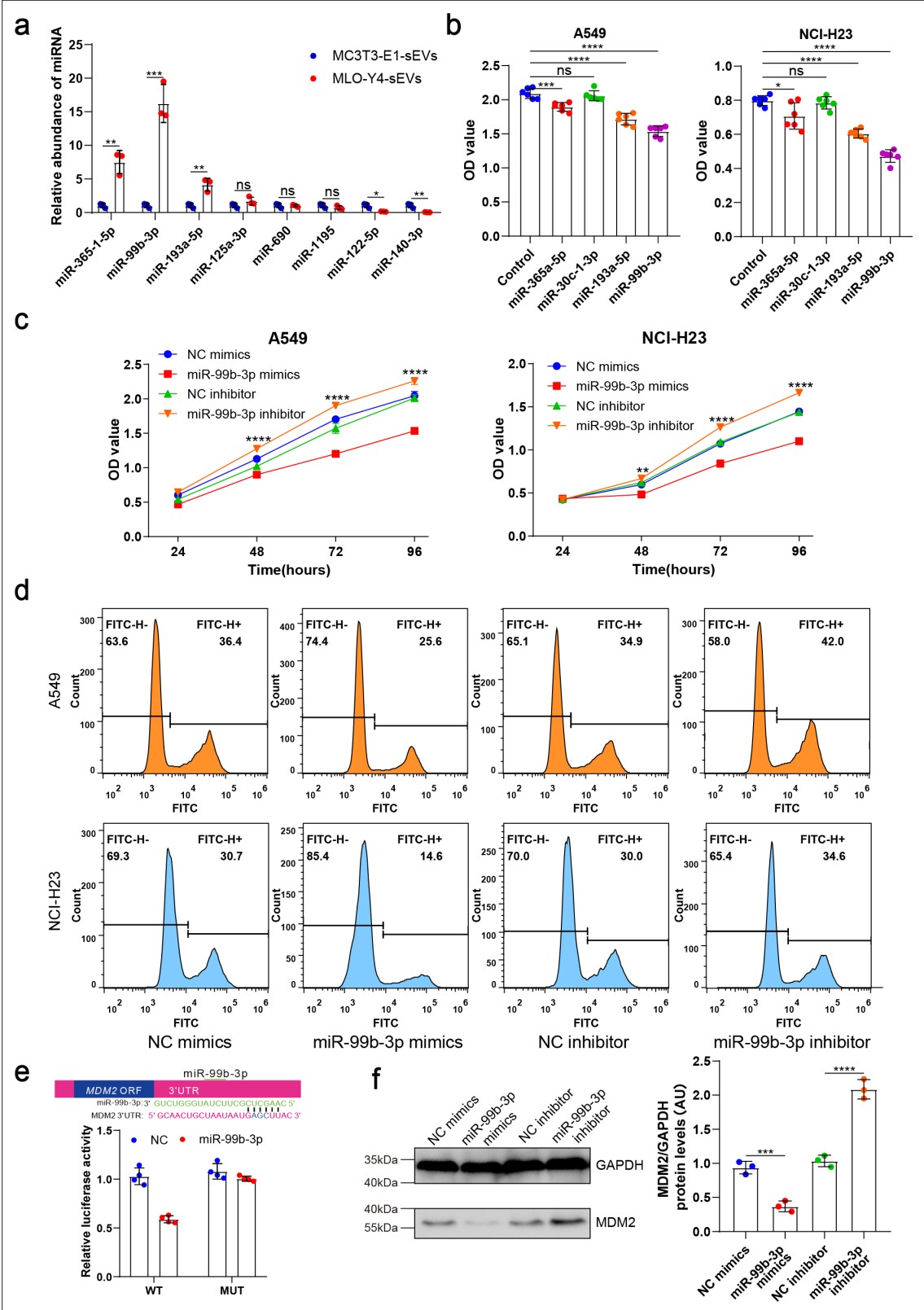

**Figure 3.** Osteocyte small extracellular vesicles (sEVs) miR-99b-3p inhibited the proliferation of non-small cell lung cancer (NSCLC) cells by directly targeting murine double minute 2 (MDM2). (**a**) Comparison of relative miR-365-1-5p, miR-99b-3p, miR-193a-5p, miR-125a-3p, miR-690, miR-1195, miR-122-5p, miR-140-3p content between MLO-Y4-sEVs and MC3T3-E1-sEVs by real-time quantitative polymerase chain reaction (qRT-PCR). n=3, Student's two-sided unpaired t test. (**b**) NSCLC cells were transfected with miR-365a-5p, miR-30c-1-3p, miR-193a-5p, miR-99b-3p mimics, or NC (negative control)

*Figure 3 continued on next page*

*Figure 3 continued*

in Dulbecco's modified eagle medium (DMEM). 48 hr later, Cell Counting Kit-8 (CCK-8) assays were performed. n=6, one-way analysis of variance with Turkey's multiple comparisons test. (**c, d**) NSCLC cells were transfected with miR-99b-3p mimics, NC mimics, miR-99b-3p inhibitor, and NC inhibitor. 48 hr later, EdU flow cytometry were performed. CCK-8 assays were performed at different points in time after transfection. n=6, two-way analysis of variance with Sidak's multiple comparisons test. (**e**) Schematic diagram of putative miR-99b-3p binding sites in the MDM2 3'-UTR. Green letters denote mutation sites. Relative luciferase activities of wild-type (WT) or mutant (MUT) MDM2 3'-UTRs were determined in A549 cells, which were co-transfected with the miR-99b-3p mimics or negative control. Luciferase activity was normalized using Renilla. n=4, Student's two-sided unpaired t test. (**f**) MDM2 protein in A549 cells was analyzed by western blot 72 hr after transfection. n=3; one-way analysis of variance with Turkey's multiple comparisons test. Error bars represent Mean ± SD. No significance (ns) $p>0.05$, *$p<0.05$, **$p<0.01$, ***$p<0.001$, ****$p<0.0001$.

The online version of this article includes the following source data and figure supplement(s) for figure 3:

**Source data 1.** Original table sources for quantification of *Figure 3* plots.

**Source data 2.** Original western blot images for *Figure 3f*.

**Source data 3.** Original western blot images for *Figure 3f* with highlighted bands and sample labels.

**Figure supplement 1.** The micro-RNA (miRNA) profiles of MLO-Y4-sEVs and MC3T3-E1-sEVs and distribution of miR-99b-3p in bone metastatic tissues of non-small cell lung cancer (NSCLC) patients.

**Figure supplement 1—source data 1.** Original table sources for quantification of *Figure 3—figure supplement 1* plots.

the biological progression of NSCLC cells in a bone microenvironment by direct tibial implantation of murine Lewis lung carcinoma (LLC) cells expressing green fluorescent protein (GFP). These mice were subsequently randomized into tumor-bearing only (control), zoledronic acid (ZA), treadmill running, and treadmill running combined with ZA (treadmill running+ZA) groups. Moderate exercise was provided by treadmill running for mice, which were involuntarily placed on a 12-lane rodent treadmill for 30 min per day at a speed of 15 cm/s for at least 5 days per week, and were euthanized after 4 weeks. Biophotonic imaging showed that tumor progression in the three treated groups was significantly slower than that in the control group, among which, the treadmill running+ZA group suppressed the growth of tumors in the tibia most effectively (*Figure 5a*). Tumor growth inhibition by the ZA and treadmill running groups was similar, with no statistical difference. Ki-67 protein has been widely used as a proliferation marker for tumor cells. In this study, immunofluorescence assessment results of Ki-67 and CCND3 were largely consistent with those of biophotonic imaging (*Figure 5b*, *Figure 5—figure supplement 1*). The Ki-67 and CCND3 expression level was lowest in the treadmill running+ZA group lung cancer cells in the tibia, which confirmed that in the treadmill running+ZA group, tumor growth in the tibia was suppressed most effectively. The micro-CT analysis showed that treadmill running and/or ZA protected bone integrity and delayed bone destruction in the presence of bone metastasis in NSCLC, among which, the treadmill running+ZA group displayed the strongest effect, specifically reflected in the BV/TV, Tb.Th, and Ct.Th (*Figure 5c–e*). The protective effect of bone integrity in the ZA and treadmill running groups was similar, with no statistical difference. Taken together, these results demonstrated that moderate exercise significantly suppressed bone metastasis progression of NSCLC. Furthermore, moderate exercise combined with ZA displayed additive effects.

## Exercise preconditioning effectively suppressed bone metastasis progression of NSCLC

Physical exercise offers therapeutic potentials for bone metastasis. However, it remains mostly unknown whether and how exercise preconditioning affects bone metastasis progression. In this study, we examined the effects of preconditioning on bone metastasis progression in mature adult mice using treadmill running. Male, 8-week-old C57BL/6J mice were subjected to 4 weeks of treadmill exercise followed by tibial implantation of NSCLC cells. Mice were randomized into tumor-bearing only (control), treadmill running before and after tibial implantation of NSCLC cells (exercise preconditioning), and treadmill running after tibial implantation of NSCLC cells (treadmill running) groups (*Figure 6—figure supplement 1a*). The biophotonic imaging showed that tumor growth in the two exercise groups was significantly slower than that in the control group, with the exercise preconditioning group suppressing the tumor growth more significantly effectively than treadmill running only after tibial implantation of NSCLC cells, which were consistent in C57BL/6J and nude mice (*Figure 6a*). Regarding the proliferation marker, the Ki-67 and CCND3 immunofluorescence assessment results were largely consistent with those of the biophotonic imaging (*Figure 6b*, *Figure 6—figure supplement*

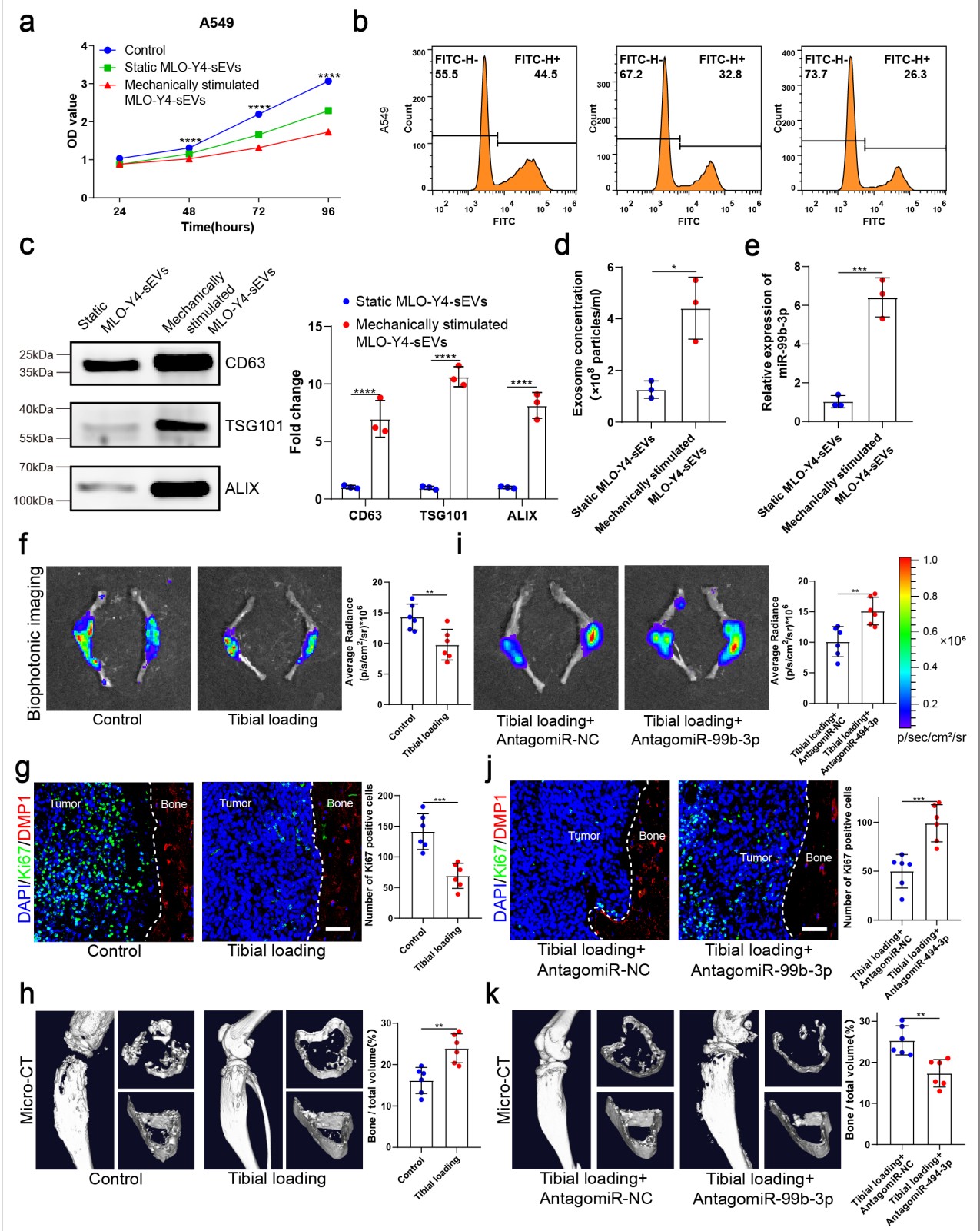

**Figure 4.** Mechanical stimulation increased the release of small extracellular vesicles (sEVs) from osteocytes and enhanced inhibitory effect of osteocytes on the proliferation of non-small cell lung cancer (NSCLC) cells. (**a**) MLO-Y4 cells were mechanically stimulated by stretching at 12% strain at a 1.25 Hz frequency for 12 hr, and sEVs were extracted from the conditioned medium. Cell Counting Kit-8 (CCK-8) assays were performed to evaluate the effect of sEVs on the proliferation of A549 cells. n=4, two-way analysis of variance with Sidak's multiple comparisons test. (**b**) EdU flow cytometry

*Figure 4 continued on next page*

*Figure 4 continued*

were performed to evaluate the effect of sEVs on the proliferation of A549 cells. (**c**) Western blot analysis of the typical sEV markers CD63, TSG101, and Alix. n=3, Student's two-sided unpaired t test. (**d**) Concentration of sEVs was detected by NanoSight analysis. n=3, Student's two-sided unpaired t test. (**e**) Mechanical loading was applied to the tibiae with bone metastasis via direct implantation of GFP-LLC into tibia of mouse. Real-time quantitative polymerase chain reaction (qRT-PCR) identification of miR-99b-3p in sEVs. n=3, Student's two-sided unpaired t test. An intraosseous model of bone metastasis was used via direct implantation of murine Lewis lung carcinoma (LLC) cells expressing green fluorescent protein (GFP) into tibia of mouse. Mice were subsequently randomized into tumor-bearing only (control), tibial loading, tibial loading+AntagomiR-NC, and tibial loading+AntagomiR-99b-3p groups. Four weeks later, mice were sacrificed. (**f, i**) Representative biophotonic images and quantification of fluorescence signal in hindlimb bones of mice. n=6, Student's two-sided unpaired t test. (**g, j**) Representative immunofluorescence staining images of Ki-67 (green), DMP1 (red), and 4',6-diamidino-2-phenylindole (DAPI) (blue) in the tibia of mice with bone metastases and quantification of the number of Ki-67-positive cells in the images. The dashed line indicates the boundary between bone and tumor. Scale bars: 50 µm. n=6, Student's two-sided unpaired t test. (**h, k**) Representative micro-computed tomography (micro-CT) imaging and quantitation of the tibia (bone/total volume) in mice. n=6, Student's two-sided unpaired t test. Error bars represent Mean ± SD. No significance (ns) p>0.05, *p<0.05, **p<0.01, ***p<0.001, ****p<0.0001.

The online version of this article includes the following source data and figure supplement(s) for figure 4:

**Source data 1.** Original table sources for quantification of *Figure 4* plots.

**Source data 2.** Original western blot images for *Figure 4c*.

**Source data 3.** Original western blot images for *Figure 4c* with highlighted bands and sample labels.

**Figure supplement 1.** Mechanical loading increased the release of small extracellular vesicles (sEVs) from osteocytes and enhanced inhibitory effect of osteocytes on the proliferation of non-small cell lung cancer (NSCLC) cells.

**Figure supplement 1—source data 1.** Original table sources for quantification of *Figure 4—figure supplement 1* plots.

*1b*). The Ki-67 and CCND3 expression level in NSCLC cells in the exercise preconditioning group was much significantly lower than that in the other groups, which indicated that exercise preconditioning could effectively inhibit tumor growth. The micro-CT analysis showed that the exercise protected bone integrity and delayed bone destruction in the presence of bone metastasis in NSCLC, with the exercise preconditioning group having a stronger effect than the other group, specifically reflected in the BV/TV, Tb.Th, and Ct.Th (*Figure 6c–e*). The micro-CT analyses of C57BL/6 mice and nude mice were consistent. Therefore, these results suggested that maintaining exercise habits may effectively prevent bone metastasis of NSCLC.

## Discussion

Although many preclinical and clinical studies have shown that exercise can inhibit bone metastasis progression (*Ligibel et al., 2022*; *Weller et al., 2021*), the mechanism remains poorly understood. Clarifying the mechanism of action is essential to determine optimal exercise programs, design appropriate combination therapy with standard therapy regimens, or identify specific targets for new treatment. In this study, we found that NSCLC cells adjacent to bone tissue had a much lower proliferative capacity than the surrounding tumor cells, thus we hypothesized that bone tissues release some substance that inhibits tumor growth and sustains tumor dormancy in bone metastasis. Next, we demonstrated that osteocytes inhibit NSCLC cell proliferation by releasing sEVs containing tumor suppressor miRNAs, such as miR-99b-3p. Meanwhile, both in vitro and in vivo experiments demonstrated that mechanical loading stimulated osteocytes to release more sEVs and miR-99b-3p to delay the progression of bone metastasis of NSCLC. In addition, we showed that bone metastasis progression of NSCLC was inhibited by moderate exercise, and combinations with ZA had additive effects. Furthermore, exercise preconditioning effectively suppressed bone metastasis progression of NSCLC. Taken together, our work elucidated a mechanism for the clinical observation that bone metastasis was inhibited by moderate exercise.

In bone metastasis, only a limited number of disseminated tumor cells survive following bone colonization, and these are retained in a dormant state for prolonged periods through engagement in specialized endosteal niches before becoming reactivated to form overt metastases (*Zhang et al., 2019*). Currently, molecular mechanisms for tumor cell dormancy in the endosteal niche remain poorly understood, representing both a challenge and an opportunity for therapeutic targeting (*Clézardin et al., 2021*). Bone metastases occur from a complex interplay between cancer cells and bone cells. Interestingly, here we found that NSCLC cells adjacent to bone tissue have a much lower proliferative

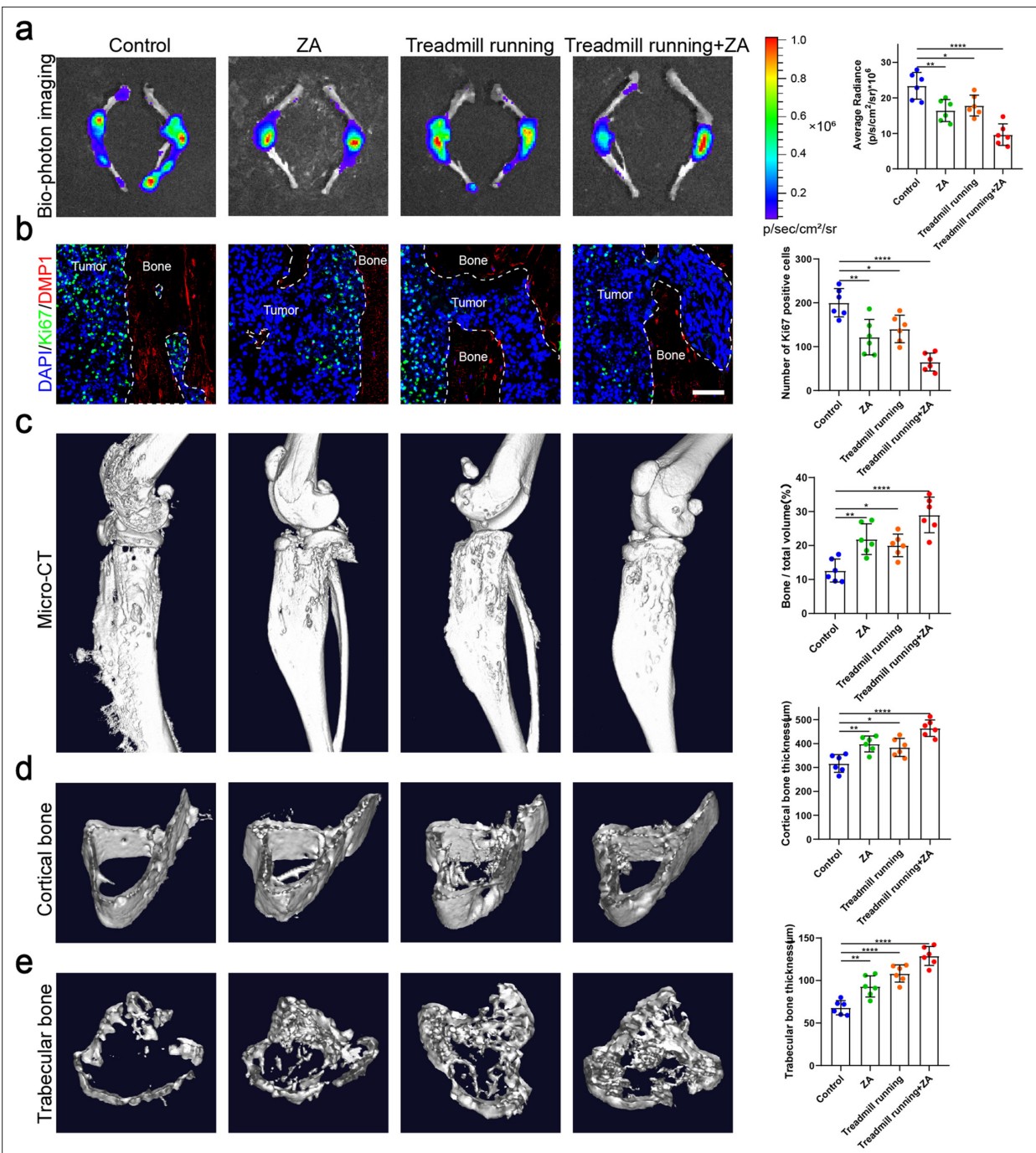

**Figure 5.** Moderate exercise combined with zoledronic acid (ZA) effectively suppressed the progression of bone metastasis of non-small cell lung cancer (NSCLC). An intraosseous model of bone metastasis was used via direct implantation of murine Lewis lung carcinoma (LLC) cells expressing green fluorescent protein (GFP) into tibia of mouse. Mice were subsequently randomized into tumor-bearing only (control), ZA, treadmill running, and treadmill running combined ZA (treadmill running+ZA) groups. Four weeks later, mice were sacrificed. (**a**) Representative biophotonic images and quantification of fluorescence signal in hindlimb bones of mice. n=6, one-way analysis of variance with Turkey's multiple comparisons test. (**b**) Representative immunofluorescence staining images of Ki-67 (green), DMP1 (red), and 4',6-diamidino-2-phenylindole (DAPI) (blue) in the tibia of mice with bone metastases and quantification of the number of Ki-67-positive cells in the images. The dashed line indicates the boundary between bone and tumor. Scale bar: 50 μm. n=6, one-way analysis of variance with Turkey's multiple comparisons test. (**c–e**) Representative micro-computed tomography (micro-CT) imaging and quantitation of the tibia (bone/total volume, cortical bone thickness, trabecular bone thickness) in mice. n=6, one-way analysis of variance with Turkey's multiple comparisons test. Error bars represent Mean ± SD. No significance (ns) p>0.05, *p<0.05, **p<0.01, ***p<0.001, ****p<0.0001.

*Figure 5 continued on next page*

*Figure 5 continued*

The online version of this article includes the following source data and figure supplement(s) for figure 5:

**Source data 1.** Original table sources for quantification of *Figure 5* plots.

**Figure supplement 1.** Moderate exercise combined with zoledronic acid effectively suppressed the progression of bone metastasis of non-small cell lung cancer (NSCLC).

**Figure supplement 1—source data 1.** Original table sources for quantification of *Figure 5—figure supplement 1* plots.

capacity than the surrounding tumor cells. Therefore, we hypothesized that bone tissues release some substance that inhibits tumor growth and sustains NSCLC cell dormancy in bone metastasis.

Osteoclast and osteoblast functions in bone metastases have been well characterized; however, the role of osteocytes needs to be elucidated (*Atkinson and Delgado-Calle, 2019*). Osteocytes act as mechanosensors to control responses to mechanical loading of the skeleton. For bone metastases, osteocytes may play a pivotal role in the interaction between mechanical loading and SREs. Here, we demonstrated that it is osteocytes that inhibit NSCLC cell proliferation in the endosteal niche, but not osteoblasts. Furthermore, we demonstrated that osteocytes inhibit NSCLC cell proliferation by releasing sEVs containing tumor suppressor miRNAs, such as miR-99b-3p. Recent studies showed that miR-99b-3p functions as a potential tumor suppressor in oral squamous cell carcinoma and gastric cancer (*Chang et al., 2019*; *He et al., 2015*). We found that osteocyte sEVs miR-99b-3p inhibited NSCLC cell proliferation by directly targeting MDM2. As a negative regulator of the p53, the most extensively studied human tumor suppressor, MDM2 has been shown to exert oncogenic activity (*Karni-Schmidt et al., 2016*). To date, three regions of interaction between MDM2 and p53 have been identified, by which MDM2 inhibits the binding of p53 to the transcriptional machinery. MDM2 is significantly highly expressed in lung adenocarcinoma tissues compared with adjacent tissues, and is regarded as a proto-oncogene (*Chen et al., 2022*; *Ni et al., 2021*). Our findings identified MDM2 as a direct target of miR-99b-3p, emphasizing the crucial role of miRNA in bone metastases. Interestingly, the miR-99b-3p content in osteocyte sEVs increased dramatically after mechanical loading, suggesting that miR-99b-3p may play a role in the inhibition of bone metastases by exercise.

SREs resulting from bone metastasis are a major clinical problem that significantly harm patient outcomes, and are commonly considered an incurable consequence of terminal malignancies. Bone metastases from lung cancer represent the majority of cases, followed by prostate cancer and breast cancer (*Ryan et al., 2022*). A recent animal study showed that in vivo exercise was protective against metastatic bone loss and inhibited breast cancer progression in bone (*Wang et al., 2021*). However, there was no report about the effect of exercise on bone metastasis of lung cancer until recently. In the present study, we showed that bone metastasis progression in lung cancer was inhibited by exercise. More interestingly, exercise preconditioning effectively suppressed bone metastasis progression of NSCLC, suggesting that maintaining exercise habits affords protection against bone metastasis. Bisphosphonates are the standard of care in patients with bone metastases. Our study demonstrated that moderate exercise and ZA have additive effects.

Exercise is quite complex because it involves multiple organs of the body. The bone can sense the mechanical stress stimulus in the process of exercise. Using in vivo models, mostly breast cancer bone metastasis models, several studies suggested that mechanical stimulation of the tibia can contribute to inhibiting bone metastatic colonization and cancer progression in bone (*Fan et al., 2020*; *Wang et al., 2021*). Tibial compression has been used to stimulate bone formation and study the interplay between mechanical stimuli and SREs. One study showed that moderate tibial loading, but not overloading, protected adult murine bone from destruction by metastasized breast cancer (*Wang et al., 2021*). Consistent with this, our work showed that mechanical loading of the tibia inhibited bone metastasis progression of NSCLC. Although many preclinical and clinical studies have shown that exercise can inhibit cancer progression, the ideal mechanism of implementation remains to be determined. The prescription of exercise for patients with cancer requires further research, including exercise type, frequency, duration, intensity, and whether it is supervised (*Cavalheri and Granger, 2020*; *Stout et al., 2021*). Exercise has been underutilized in patients with advanced cancer despite the benefits, particularly for those with bone metastases because of concerns over skeletal complications (*Cheville et al., 2011*; *Ten Tusscher et al., 2020*). Multiple studies showed that individuals with NSCLC were engaged in significantly less physical activity than similar-aged healthy individuals

(*Granger et al., 2014a*; *Granger et al., 2014b*). Exercise models such as running are more realistic for healthy people, but mechanical loading such as tibial compression provides for isolating the impacts of mechanical stimuli on bone tissue in the context of SREs, which are more effective and safer for patients with advanced cancer. Previous studies have shown that either exercise or mechanical loading can lead to increased bone mass, which may be another reason for the attenuated erosion of NSCLC cells and deserves our further exploration (*Robling et al., 2006*).

In conclusion, our study provides new insights into understanding the mechanism underlying exercise-afforded protection against bone metastasis progression. Bone metastasis progression of NSCLC was inhibited by moderate exercise. Osteocytes, sensing mechanical stimulation generated by exercise, inhibit the proliferation of NSCLC cells and sustain their dormancy by releasing sEVs containing tumor-suppressive miRNAs (*Figure 7*). Understanding the mechanism underlying tumor dormancy may provide the foundation for the development of strategies to prevent dormant bone metastasis from emerging as overt bone metastasis.

# Materials and methods

**Key resources table**

| Reagent type (species) or resource | Designation | Source or reference | Identifiers | Additional information |
|---|---|---|---|---|
| Cell line (*Homo sapiens*) | A549 | ATCC | CVCL_0023 | |
| Cell line (*H. sapiens*) | NCI-H23 | ATCC | CVCL_1547 | |
| Cell line (*Mus musculus*) | MLO-Y4 | ATCC | CVCL_M098 | |
| Cell line (*M. musculus*) | MLO-A5 | ATCC | CVCL_0P24 | |
| Cell line (*M. musculus*) | MC3T3-E1 | ATCC | CVCL_0409 | |
| Cell line (*M. musculus*) | LLC | ATCC | CVCL_4358 | |
| Cell line (*H. sapiens*) | NCM460 | ATCC | CVCL_0460 | |
| Antibody | Anti-CD63 (Mouse monoclonal) | Abcam | ab193349 | 1:1000 |
| Antibody | Anti-TSG101 (Mouse monoclonal) | Abcam | ab83 | 1:1000 |
| Antibody | Anti-ALIX (Rabbit polyclonal) | Abcam | ab76608 | 1:1000 |
| Antibody | β-Actin (Rabbit monoclonal) | Cell Signaling Technology | 4970S | 1:4000 |
| Antibody | GAPDH (Rabbit monoclonal) | Cell Signaling Technology | 2118 | 1:5000 |
| Antibody | MDM2 (Rabbit polyclonal) | Proteintech | 66511-1-IG | 1:1000 |
| Antibody | Ki-67 (Rabbit monoclonal) | Cell Signaling Technology | 9129 | 1:200 |
| Antibody | DMP1 (Sheep polyclonal) | R&D Systems | AF4386 | 1:100 |
| Antibody | GFP (Mouse monoclonal) | ABclonal | AE012 | 1:100 |
| Antibody | CCND3 (Rabbit polyclonal) | Bioss | bs-0660R | 1:100 |
| Sequence-based reagent | Cel-miR-39-3p_F | Vazyme | qRT-PCR Forward primer | 5'-GCGTCACCG GGTGTAAATC-3' |
| Sequence-based reagent | mmu-miR-365-1-5p_F | Vazyme | qRT-PCR Forward primer | 5'-AGGGACTTT TGGGGGCA-3' |
| Sequence-based reagent | mmu-miR-99b-3p_F | Vazyme | qRT-PCR Forward primer | 5'-GCGCAAGCT CGTGTCTGTG-3' |

*Continued on next page*

*Continued*

| Reagent type (species) or resource | Designation | Source or reference | Identifiers | Additional information |
|---|---|---|---|---|
| Sequence-based reagent | mmu-miR-193a-5p_F | Vazyme | qRT-PCR Forward primer | 5'-TGGGTCTT TGCGGGCA-3' |
| Sequence-based reagent | mmu-miR-125a-3p_F | Vazyme | qRT-PCR Forward primer | 5'-CGCGACAG GTGAGGTTCTTG-3' |
| Sequence-based reagent | mmu-miR-690_F | Vazyme | qRT-PCR Forward primer | 5'-GCGAAAGG CTAGGCTCACA-3' |
| Sequence-based reagent | mmu-miR-1195_F | Vazyme | qRT-PCR Forward primer | 5'-GTGAGTTCG AGGCCAGCC-3' |
| Sequence-based reagent | mmu-miR-122-5p_F | Vazyme | qRT-PCR Forward primer | 5'-CGCGTGGAGT GTGACAATGG-3' |
| Sequence-based reagent | mmu-miR-140-3p_F | Vazyme | qRT-PCR Forward primer | 5'-GCGCGTACC ACAGGGTAGAA-3' |
| Sequence-based reagent | Reverse primer of all miRNAs | Vazyme | qRT-PCR Reverse primer | 5'-AGTGCAGGGT CCGAGGTATT-3' |
| Commercial assay or kit | PKH67 Green Fluorescent Cell Linker Kit | Sigma-Aldrich | PKH67GL | |
| Commercial assay or kit | Cell-Light EdU Apollo488 In Vitro Kit | Ribobio | C10310-3 | |
| Commercial assay or kit | miRCURY LNA miRNA Detection Kit | Exiqon | 339115 | |
| Chemical compound, drug | GW4869 | Sigma-Aldrich | D1692 | |
| Software, algorithm | GraphPad Prism | GraphPad Software | Version 6.0 RRID:SCR_002798 | |

## Patients

Lung cancer patients with a diagnosis of bone metastases (six patients, radiologically confirmed) were recruited by the Third Affiliated Hospital of Southern Medical University (Guangzhou, China) from January 2020 to January 2021. All patients were fully informed of the study, and they provided written consent prior to participation. Surgically resected bone metastatic tumor tissue was collected from all patients. The obtained bone metastatic tumor tissue was trimmed to an appropriate size and rapidly transferred to formalin solution for storage until processing. We confirm that our study was compliant with the 'Guidance of the Ministry of Science and Technology (MOST) for the Review and Approval of Human Genetic Resources'. Ethics approval was granted by the Human Research Ethics Committee of the Third Affiliated Hospital of Southern Medical University (2019-LS-16). The study was conducted in accordance with the principles and guidelines of The Declaration of Helsinki.

## Cell lines

A549, LLC, NCM460, MLO-A5, and MLO-Y4 cells were purchased from the American Type Culture Collection (Rockville, MD, USA) and grown in Dulbecco's modified eagle medium (DMEM) (Invitrogen, Waltham, MA, USA) containing 10% FBS and 1% penicillin and streptomycin. MC3T3-E1 cells were purchased from the American Type Culture Collection and were cultured in α Minimum Essential Medium (Invitrogen) with 10% FBS and 1% penicillin and streptomycin. All cell lines were mycoplasma negative and authenticated by ATCC STR profiling.

## Animal models

All animal experimental protocols were approved by the Animal Care and Use Committee of the Southern Medical University, Guangzhou, China (SMUL2020141). C57BL/6 and nude athymic mice were obtained from the Experimental Animal Center of the Southern Medical University in Guangzhou, China. Animal care was in accordance with the guidelines of the US National Institutes of Health and the Chinese National Institute of Health. All surgical interventions were performed under anesthesia with a mixture of 13.3% urethane and 0.5% chloralose (0.65 ml/100 g body weight). All efforts

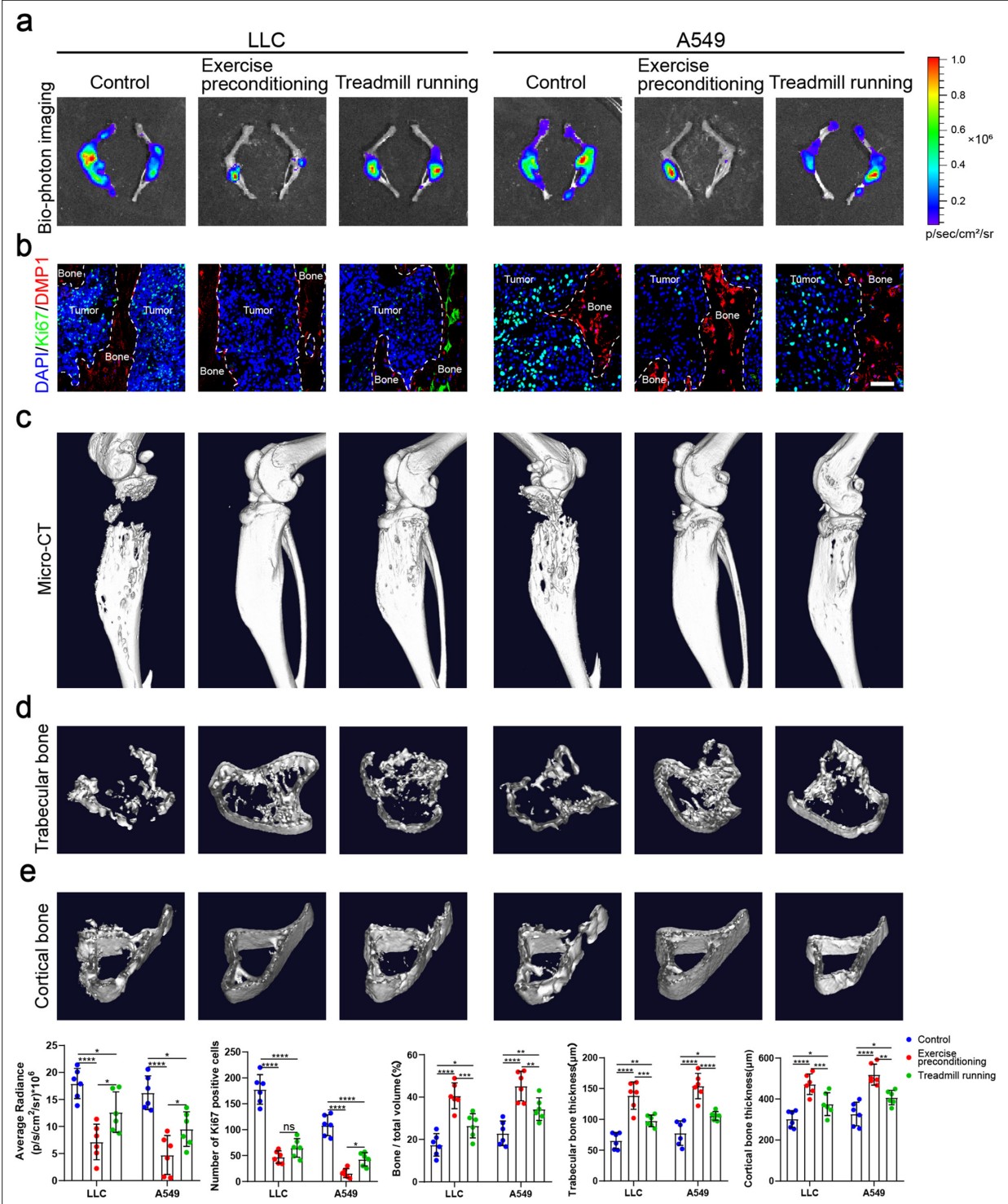

**Figure 6.** Exercise preconditioning effectively suppressed the progression of bone metastasis of non-small cell lung cancer (NSCLC). An intraosseous model of bone metastasis was used via direct implantation of NSCLC cells (Lewis lung carcinoma [LLC] and A549) into tibia of mouse. Mice were subsequently randomized into tumor-bearing only (control), treadmill running before and after implantation of NSCLC cells into tibia (exercise preconditioning), and treadmill running after implantation of NSCLC cells into tibia (treadmill running) groups. For exercise preconditioning group, mice were subjected to 4 weeks of treadmill exercise before implantation of NSCLC cells into tibia of mouse. Four weeks after implantation, mice were sacrificed. (**a**) Representative biophotonic images and quantification of fluorescence signal in hindlimb bones of mice. n=6, one-way analysis of variance with Turkey's multiple comparisons test. (**b**) Representative immunofluorescence staining images of Ki-67 (green), DMP1 (red), and 4',6-diamidino-2-phenylindole (DAPI) (blue) in the tibia of mice with bone metastases and quantification of the number of Ki-67-positive cells in the images. The dashed

*Figure 6 continued on next page*

*Figure 6 continued*

line indicates the boundary between bone and tumor. Scale bar: 50 µm. n=6, one-way analysis of variance with Turkey's multiple comparisons test. (**c–e**) Representative micro-computed tomography (micro-CT) imaging and quantitation of the tibia (bone/total volume, cortical bone thickness, trabecular bone thickness) in mice. n=6, one-way analysis of variance with Turkey's multiple comparisons test. Error bars represent Mean ± SD. No significance (ns) $p>0.05$, *$p<0.05$, **$p<0.01$, ***$p<0.001$, ****$p<0.0001$.

The online version of this article includes the following source data and figure supplement(s) for figure 6:

**Source data 1.** Original table sources for quantification of *Figure 6* plots.

**Figure supplement 1.** Exercise preconditioning effectively suppressed the progression of bone metastasis of non-small cell lung cancer (NSCLC).

**Figure supplement 1—source data 1.** Original table sources for quantification of *Figure 6—figure supplement 1* plots.

were made to reduce the number of animals used and to minimize animal discomfort. Animals were randomly grouped in all experiments, and the names of the groups were concealed until statistical analysis was completed.

For in situ studies, $25\times10^4$ LLC cells or A549 cells were injected into the tibiae of 8-week-old C57BL/6 mice or nude mice. In brief, $25\times10^4$ LLC cells and A549 cells suspended in phosphate-buffered saline (PBS) were injected into the tibia of C57BL/6 mice and nude mice, respectively. Mice were euthanized on day 21 for analyses, unless otherwise indicated. For stress-induced experiments, after 2 days of recovery, tibiae loading was performed daily on the tumor-inoculated tibiae. Using an ElectroForce device (Bose, Framingham, MA, USA), the tibiae of the mask-anesthetized mice were given daily loads of 1 N (peak-to-peak) at 2 Hz for 10 min, 5 days per week for 4 weeks. The tibias of a subset of mice were injected with AntagomiR-NC or AntagomiR-99b-3p (GenePharma, Shanghai, China, 3 nmol per tibia, once a week for 3 weeks) after 2 days of recovery.

For ZA experiments, starting on day 3 after lung cancer cell transplantation, mice were treated with subcutaneous injection of 100 µg/kg ZA (Novartis, Basel, Switzerland) or saline (other groups) every 3 days for 3 weeks, after which they were euthanized. For exercise experiments, exercised mice were involuntarily placed on a 12-lane rodent treadmill for 30 min per day at a speed of 15 cm/s for at least 5 days per week unless otherwise stated. During the 3 days prior to euthanasia, the exercising mice underwent obligatory exercise, regardless of how many consecutive days they had previously exercised. Two days after surgery, all mice except the control group began moderate exercise by treadmill running, whereby they were involuntarily placed on a 12-lane rodent treadmill for 30 min per

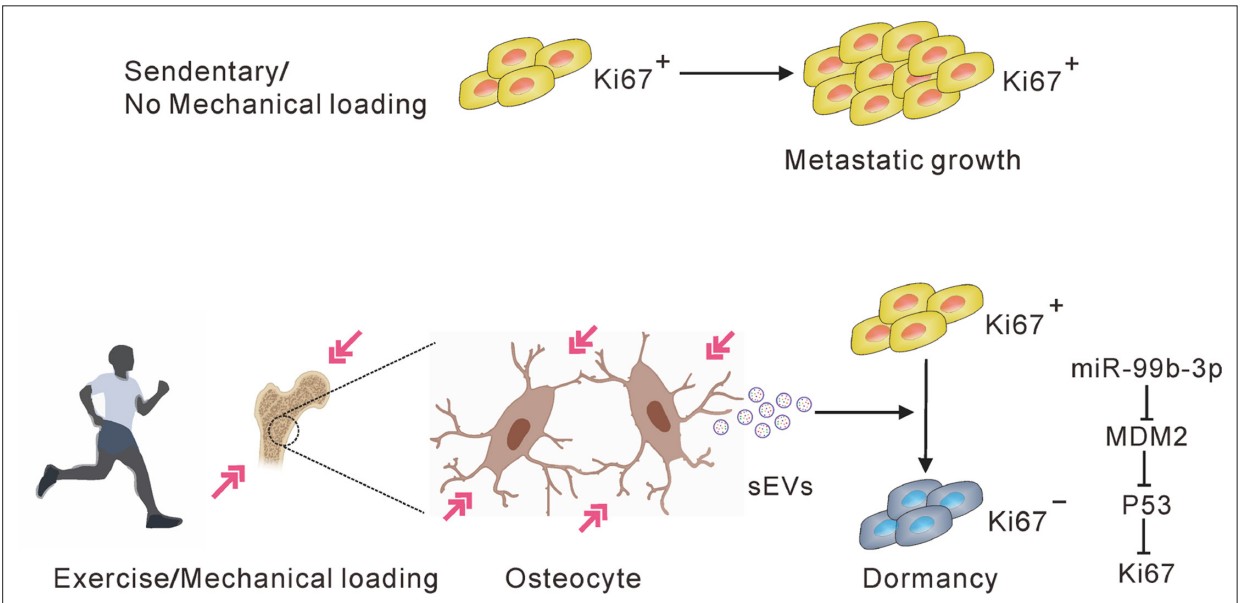

**Figure 7.** A hypothetical model depicting osteocytes induces and maintains tumor dormancy in bone metastasis of non-small cell lung cancer (NSCLC). Osteocytes, sensing mechanical stimulation generated by exercise or mechanical loading, inhibit the proliferation and sustain the dormancy of NSCLC cells by releasing small extracellular vesicles (sEVs) with tumor suppressor micro-RNAs (miRNAs), such as miR-99b-3p, which inhibits the proliferation of NSCLC cells by directly targeting murine double minute 2 (MDM2).

day at a speed of 15 cm/s for at least 5 days per week, and were euthanized after 4 weeks. For exercise preconditioning, mice were additionally subjected to 4 weeks of treadmill exercise prior to tibial implantation of NSCLC cells. In the matched experiment, the control mice were placed on a stationary (0 cm/s) treadmill for the same time duration as the experimental mice.

We examined the effects of preconditioning on bone metastasis progression in mature adult mice using treadmill running. Male 8-week-old C57BL/6J mice were subjected to 4 weeks of treadmill exercise followed by tibial implantation of NSCLC cells.

## Lung cancer cell preparation and in situ injection

We used clones derived from the GFP-tagged LLC cells and A549 cells. The cells were maintained in DMEM containing 10% FBS and 0.1 mg/ml penicillin-streptomycin at 37°C under 5–8% $CO_2$. Cells were passaged at 80–90% confluence and re-plated at a 1:10 dilution, which was approximately every 72 hr. Before in situ injection, cells were trypsinized at 80–90% confluence with 0.15% trypsin/ethylene diamine tetraacetic acid (EDTA) (0.5% Gibco trypsin EDTA diluted 1:2 in PBS). Cells were immediately removed from the plate with 10 ml ice-cold DMEM containing 10% FBS, pipetted into a 15 ml conical tube, and centrifuged at $200 \times g$ for 5 min. The pellet was resuspended in a 50 ml conical tube in 25 ml ice-cold PBS (without Ca or Mg) and counted. This was centrifuged to re-pellet, and finally resuspended at $25 \times 10^4$ cells/10 µl in ice-cold PBS.

The mice were anesthetized using a mixture of 13.3% urethane and 0.5% chloralose (0.65 ml/100 g body weight). Both legs were cleaned with 10% povidone/iodine swab/solution, followed by ethanol, repeated twice. The skin was wetted with 70% ethanol to increase visibility of the underlying patellar ligament, which should be visible as a distinct, thick, white line. While firmly grasping the ankle/leg of the mouse, a 28 g ½ needle was inserted under the patella, through the middle of patellar ligament, and into the anterior intercondylar area in the top of the tibia. A gentle, lateral movement of the needle was applied to ensure it was in tibia and passed through the growth plate. The plunger was slowly depressed to inject 10 µl of cell solution. After 3 min, the needle was slowly extracted. The treated mice were kept on a heating pad until revived.

## Mechanical stress stimulation of MLO-Y4 cells

MLO-Y4 cells were transferred to a Biofex six-well plate (Flexcell, Burlington, NC, USA) and allowed to adhere to the membrane until they attained 90% confluence. The cells were subjected to sinusoidal stretching (Flexcell) at 12% strain and 1.25 Hz frequency for 12 hr. Experiments were performed at 37°C under 5% $CO_2$. Following 48 hr of stretching, the CM (or control medium) was collected for sEVs extraction.

## Biophoton imaging and micro-CT imaging

Tumor-bearing mice were euthanized 30 days after tumor cell injection, and their hindlimb bones were imaged by biophotonics. Fluorescence imaging for GFP-labeled tumor cells in these bones was performed using the optical imaging system, Fx Pro (Bruker BioSpin MRI GmbH, Ettlingen, Germany). For anatomical orientation, a white light/grayscale picture was performed and used with the fluorescent signal (Cy3.5: excitation = 570 nm; emission = 620 nm). For data analysis, molecular imaging software 7.1.3 (Bruker BioSpin MRI GmbH) was used. Regions of interest (ROIs) were defined around the bone and analyzed compared to the background of PBS injection to calculate the signal to noise ratio.

Micro-CT was performed using the Skyscan 1172 (Bruker-MicroCT, Kontich, Belgium). The X-ray tube settings were 80 kV and 500 µA and images were acquired at the highest resolution without charge-coupled device binning, resulting in a voxel size of 9.44 µm. A 0.66° rotation step through a 195° angular range with 6500 ms exposure was used. After processing with a three-dimensional Gaussian filter to reduce noise, the ROIs were manually segmented such that they corresponded to the cortical and trabecular bone regions. Cortical ROIs comprised 470 µm thick regions (50 slices, 9.4 µm thickness per slice) located 3 mm distal to the growth plate, while 300 µm thick trabecular sections began 0.27 mm distal to the growth plate.

## Histological analysis, immunohistochemistry, and immunofluorescence staining

Bone samples were fixed in 4% paraformaldehyde in PBS and decalcified in a 10% EDTA solution. They were subsequently dehydrated through a series of graded alcohols, cleared in xylene, and embedded in paraffin. Sagittal sections of 2–5 μm were prepared for histological analyses. Following deparaffinization and rehydration, H&E staining were performed on the tissue sections. The tissue sections were imaged using an FSX100 microscope (Olympus, Tokyo, Japan).

For immunohistochemistry, sections were incubated in 10 mM citrate buffer (pH 6.0) for 30 min at 90°C or treated with 200 mg/ml proteinase K (Sigma-Aldrich, St. Louis, MO, USA) for 10 min at 37°C to unmask antigen. Subsequently, the sections were treated with 3% hydrogen peroxide for 15 min. Thereafter, the sections were permeabilized with 0.1% Triton X-100 in PBS for 5 min at room temperature and blocked with 1% sheep serum at room temperature for 1 hr. We incubated the sections with a primary antibody that recognized Ki-67 (Cell Signaling Technology, Danvers, MA, USA, cat. #: 9129, 1:200) overnight at 4°C. Subsequently, we used a horse radish peroxidase-labeled secondary antibodies (1:200 in 1% bovine serum albumin [BSA], 1 hr) at 37°C. Diaminobenzidine was used as a chromogen and hematoxylin as the counterstain.

For immunofluorescence, we incubated primary antibodies that recognized Ki-67 (Cell Signaling Technology, cat. #: 9129, 1:200), DMP1 (R&D Systems, Minneapolis, MN, USA, cat. #: AF4386, 1:100), GFP (ABclonal, Wuhan, China, cat. #: AE012, 1:100), and CCND3 (Bioss, Woburn, MA, USA, cat. #: bs-0660R, 1:100) overnight at 4°C. For secondary reactions, species-matched Alexa Fluor 488- and Alexa Fluor 594-secondary antibodies were used (1:500 in 1% BSA, 1 hr) at 37°C in the dark. The secondary antibodies for immunofluorescence staining were goat anti-rabbit Alexa Fluor 488 (Invitrogen, cat, #: A-11008) and donkey anti-Sheep Alexa Fluor 594 (Invitrogen, cat. #: A-11016). The sections were mounted with 4',6-diamidino-2-phenylindole (DAPI) (Thermo Fisher, Waltham, MA, USA) before imaging and we examined more than five different microscopic images (per section) under a confocal laser scanning microscope (Olympus, Tokyo, Japan).

## In situ hybridization

In situ hybridization was performed to determine the expression of miR-99b-3p using miRCURY LNA miRNA Detection, Optimization Kit 2 (miR-99b-3p) (Exiqon, Vedbæk, Denmark), according to the manufacturer's protocol. Sections were serially cut to 4 μm thicknesses, the tissues were subjected to pepsin (1.3 mg/ml, Sigma-Aldrich) for 30 min. After washing in PBS, the slides were submerged in 99.7% ethanol and air-dried. Slides were hybridized with 40 nM miR-99b-3p in an incubation chamber at 37°C for 16–18 hr, followed by stringent washes with saline-sodium citrate buffer at 37°C. For immunodetection, slides were blocked with digoxigenin blocking reagent (Roche, Mannheim, Germany) in maleic acid buffer containing 2% sheep serum at ambient temperature for 60 min and then incubated with sheep anti-digoxigenin conjugated to ALP (diluted 1:800 in blocking reagent) at ambient temperature for 60 min, resulting in dark-blue staining. The NBT/BCIP reaction mixture was incubated within a humidified chamber at 37°C for 2 hr and the reaction was stopped with KTBT buffer, and then washed in PBS three times, dehydrated, and mounted for microscopy.

## Western blot analysis

Cells and sEVs were lysed with 2% sodium dodecyl sulfonate (SDS), 2 M urea, 10% glycerol, 10 mM Tris-HCl (pH 6.8), 10 mM dithiothreitol, and 1 mM phenylmethylsulfonyl fluoride. The lysates were separated by SDS-polyacrylamide gel electrophoresis and blocked with BSA. Following blotting onto a nitrocellulose membrane (Bio-Rad Laboratories, Hercules, CA, USA), the membrane was incubated with primary antibodies that recognized TSG101 (Abcam, Cambridge, UK, cat. #: ab83, 1:1000), CD63 (Abcam, cat. #: ab193349, 1:1,00), CD9 (Abcam, cat. #: ab223052, 1:1000), Alix (Abcam, cat. #: ab76608, 1:1000), β-actin (Cell Signaling Technology, cat. #: 4970S, 1:4000), MDM2 (Proteintech, Wuhan, China cat. #: 66511-1-IG, 1:1000), and GAPDH (Cell Signaling Technology, cat. #: 2118, 1:5000) overnight at 4°C. Secondary antibodies for western blot were anti-rabbit IgG (cat. #: 7074, dilution 1:2000) from Cell Signaling Technology and anti-mouse IgG (cat. #: A9044, dilution 1:2000) from Sigma-Aldrich (1 hr at 37°C). The membrane was analyzed using specific antibodies and visualized using an enhanced chemiluminescence 50 kit (Amersham Biosciences, Piscataway, NJ, USA).

## Transwell assay

The migration capacity of cancer cells was determined using a 24-well plate, Transwell chambers (Thermo Fisher) with an 8 μm pore size, and Matrigel (100 μg/ml). Approximately $5×10^4$ cells in 200 μl serum-free DMEM were plated in the upper chambers, and 800 μl CM was added to the lower chambers. After 48 hr, the cells that had invaded to the lower side of the membrane were stained with crystal violet. At minimum of five randomly selected images were acquired, and the mean number of stained cells was determined.

## Luciferase assays

A549 cells were cultured at $1×10^5$ cells/well in 12-well plates. The cells were co-transfected with miR-99b-3p mimic (50 nM) or miRNA mimic negative control (50 nM) and 0.2 μg psiCHECK-1-UTR. Transfection was performed using Lipofectamine 3000. After 48 hr, cells were collected and luciferase activity was determined using the DualLuciferase reporter assay system (Promega, Madison, WI, USA) with the dual luciferase assay reporter-ready luminometer (Promega). The assays were performed in triplicate.

## TEM and NanoSight tracking analysis

The morphology of the sEVs was observed by TEM. Briefly, the sEVs suspension was mixed with an equal amount of 4% paraformaldehyde. Following washing with PBS, 4% uranyl acetate was added for chemical staining of sEVs, and images were captured using a Hitachi H-7650 TEM (Hitachi, Tokyo, Japan). NanoSight tracking analysis was performed to determine the size and concentration of the isolated sEVs using the NanoSight NS500 (Malvern, Westborough, MA, USA), according to the operating instructions, without any changes. The sEVs were diluted to be within the recommended concentration range. Five 60 s videos were captured for each sample during flow mode (camera settings: slider shutter 890, slider gain 146). SEVs were diluted and loaded onto the NS500 instrument by a syringe. The videos were analyzed with the NTA 3.2 software (Malvern). All measurements were performed at room temperature.

## RNA isolation

RNA was extracted from cells using the QIAzol Lysis Reagent (QIAGEN, Hilden, Germany) according to the user guidelines. Briefly, cells were collected in a reaction tube, lysed with QIAzol, and mixed with chloroform. Following centrifugation at 12,000×$g$ for 15 min at 4°C, the upper aqueous phase was transferred to an RNeasy Mini spin column in a 2 ml collection tube and mixed with 100% ethanol. Following washing at 7500×$g$ for 5 min at 4°C, the total RNA was collected for qRT-PCR analysis. RNA was extracted from sEVs fractions using the QIAGEN miRNeasy Mini kit (QIAGEN) according to the manufacturer's instructions, with a final elution volume of 50 μl.

## miRNA sequencing

Total RNA extraction from the samples was conducted using the miRNeasy Kit (QIAGEN), according to the standard operating procedures provided by the manufacturer. The quality and integrity of the total RNA were determined using the Agilent 2100 Bioanalyzer and the RNA 6000 Nano LabChip Kit (Agilent, Santa Clara, CA, USA). A sequencing RNA library was constructed by performing a 3′-end linker, a 5′-end linker, reverse transcription, amplification, cDNA library size selection, and purification steps using total RNA. Cluster generation and first stage sequencing primer hybridization were performed on the cBot of the Illumina HiSeq sequencer from Shanghai Biotechnology (Shanghai, China), according to the corresponding procedure in the cBot User Guide. Subsequently, the sequencing reagent was prepared following the Illumina User Guide and the flow cell carrying the cluster was loaded into the machine. Single-end sequencing was performed using the single-read program. The sequencing process was controlled by the data collection software provided by Illumina that performed real-time data analyses.

## RT-PCR and qRT-PCR

RNA samples were quantitated and qualified using a NanoDrop analyzer (Thermo Fisher). Equal quantities (5 ng) of total RNA from each sample were used for cDNA synthesis using the PrimeS-criptRT reagent kit (TaKaRa, Tokyo, Japan). The reverse transcriptions of miRNAs were performed

using looped miRNA-specific RT primers for miRNAs. qRT-PCR was performed with the Step One Plus (Applied Biosystems, Carlsbad, CA, USA), using an SYBR Green I Real-Time PCR Kit (GenePharma) for miR-99b-3p. Dissociation curves were generated to ensure the specificity of each qRT-PCR. The relative expression levels of miRNAs in each sample were calculated and quantified using the $2^{-\Delta\Delta CT}$ method after normalization for expression of the positive control.

## Co-culture experiments

Well inserts with a 0.4 mm pore size filter (BD Falcon, Franklin Lake, NJ, USA) for six-well plates were used following the manufacturer's instructions. MLO-Y4 and MC3T3-E1 cells were seeded into the well inserts ($5\times10^5$ cells/well) with DMEM. LLC, A549, and MLO-Y4 cells were also seeded into six-well plates ($1\times10^6$ cells/well). After 48 hr of co-culture, cells in six-well plates were subjected to the next experiments. All co-culture experiments were conducted in DMEM with sEVs-free FBS (Life Technologies, Carlsbad, CA, USA).

## Differential ultracentrifugation of several fractions from CM and sEVs isolation

To isolate the different CM fractions, $1\times10^6$ cells were plated in a 10 cm dish (10 ml medium). After 72 hr we either separated the CM or isolated sEVs. For isolated bone marrow-derived sEVs, suspensions were obtained by flushing mouse bone marrow with medium. The supernatant was aspirated after settling, and impurities and cells from the bone marrow were removed by centrifugation ($300\times g$ for 5 min and $600\times g$ for 10 min, subsequent steps were performed as for CM). For the dissection of the CM, whole CM (10 ml for one 10 cm dish) was collected by pipetting from the 10 cm dish into 50 ml falcon tubes, centrifuged at low speed ($2000\times g$ for 20 min) to eliminate dead cells and cellular debris prior to use, and divided into two. One half was used as whole CM, concentrated with a 10 KD ultrafiltration device (Millipore, Billerica, MA, USA) at $5000\times g$ for 15 min and were transferred to 12 wells of a 96-well plate, while the remaining half was further processed. From the second half, lEVs were collected after a $10,000\times g$ centrifugation step for 1 hr, washed in 15 ml PBS, and centrifuged again at $10,000\times g$ for 1 hr. The SF was filtered through a 0.22 mm filter prior to a $100,000\times g$ centrifugation step. The SF was collected after a 80 min $100,000\times g$ centrifugation step and concentrated using a 10 KD ultrafiltration device (Millipore) at $5000\times g$ for 15 min, obtaining concentrated factor. The final $100,000\times g$ pellet (sEVs) was washed once in 15 ml PBS and resuspended in 10% EV-depleted FBS medium for the functional cell culture experiments. Each individual SF, lEVs, and sEVs fraction used were transferred to 12 wells of a 96-well plate. SEVs from 10 ml of CM were added either to 6 wells of a 96-well plate or 1 well of a 12-well plate. Media with EV-depleted serum (10% FBS) was used as a negative control in most experiments.

## CCK-8 assay

The CCK-8 was used to determine cell viability. In total, $2\times10^3$ cells were seeded in 96-well plates and allowed to adhere overnight. Following incubation with the compounds under evaluation for 24 hr, 48 hr, 72 hr, and 96 hr (the bar chart shows the results at 72 hr), 10 µl CCK-8 dye was added to each well, and the cells were incubated for 1 hr at 37°C. Subsequently, the absorbance was determined at 450 nm (BioTek Synergy-HTX, Winooski, VT, USA).

## Flow cytometry

Approximately 10,000 cells were seeded in 12-well plates on day 1. Treatment was added on day 2, and cell proliferation was examined using a fluorescence-based cell proliferation kit (Cell-Light EdU Apollo488 In Vitro Flow Cytometry Kit, Ribobio, Guangzhou, China) on day 4. Following fluorescence labeling, the ratio of the number of fluorescently labeled cells to the total number of cells was determined by flow cytometry (Beckman, Brea, CA, USA).

## SEVs labeling and cellular uptake

SEVs were labeled using the PKH67 Green Fluorescent Cell Linker Kit (Sigma-Aldrich) by following the manufacturer's protocol. The isolated sEVs diluted in PBS was added to 0.5 ml Diluent C. Subsequently, 2 µl PKH67 dye was added and incubated for 4 min at room temperature. Two milliliters of 1% BSA/PBS was added to bind excess dye. The labeled sEVs were washed at $100,000\times g$ for 70 min, and

the sEVs pellet was suspended in PBS and used for uptake experiments. We subsequently co-cultured these PKH67 sEVs with A549. After the indicated time of co-culture, we stained the A549 with DAPI (Sigma-Aldrich) and observed them by confocal microscopy.

## Statistical analysis

Experiments were usually performed with at least three independent repeats (biological replicates) to ensure the results. All data were analyzed for statistical significance using GraphPad Prism 6.0 software (GraphPad Software, San Diego, CA, USA). The p value was determined by the Student's t test for two-group, or one-way analysis of variance test for multiple group comparisons. It was considered indicative of statistical significance when $p<0.05$. All experiments were repeated a minimum of three times. Quantitative data were expressed as mean ± standard error of the mean.

## Additional information

### Funding

| Funder | Grant reference number | Author |
| --- | --- | --- |
| National Natural Science Foundation of China | 81991511 | Xiaochun Bai |
| National Natural Science Foundation of China | 81402373 | Qiancheng Song |
| Basic and Applied Basic Research Foundation of Guangdong Province | 2023A1515110054 | Jing Xie |

The funders had no role in study design, data collection and interpretation, or the decision to submit the work for publication.

### Author contributions

Jing Xie, Data curation, Funding acquisition, Investigation, Methodology, Writing - original draft, Project administration, Writing - review and editing; Yafei Xu, Resources, Investigation, Methodology, Project administration; Xuhua Liu, Data curation, Investigation, Methodology; Li Long, Investigation, Methodology; Ji Chen, Chunyan Huang, Ruixin Zhou, Investigation; Yan Shao, Methodology; Zhiqing Cai, Jiarong Leng, Formal analysis; Zhimin Zhang, Software; Xiaochun Bai, Resources, Supervision, Funding acquisition, Project administration, Writing - review and editing; Qiancheng Song, Conceptualization, Resources, Funding acquisition, Writing - original draft, Project administration, Writing - review and editing

### Author ORCIDs

Jing Xie http://orcid.org/0000-0001-5624-458X
Xiaochun Bai http://orcid.org/0000-0001-9631-4781
Qiancheng Song http://orcid.org/0000-0002-9335-5147

### Ethics

All patients were fully informed of the study, and they provided written consent prior to participation. Ethics approval was granted by the Human Research Ethics Committee of the Third Affiliated Hospital of Southern Medical University (2019-LS-16).

All animal experimental protocols were approved by the Animal Care and Use Committee of the Southern Medical University, Guangzhou, China (SMUL2020141). All surgical interventions were performed under anesthesia. All efforts were made to reduce the number of animals used and to minimize animal discomfort.

Reviewer #1 (Public Review): https://doi.org/10.7554/eLife.89613.3.sa1
Reviewer #2 (Public Review): https://doi.org/10.7554/eLife.89613.3.sa2
Author Response https://doi.org/10.7554/eLife.89613.3.sa3

## Additional files

### Supplementary files

• MDAR checklist

### Data availability

All data generated or analysed during this study are included in the manuscript and supporting files.

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
