## [Editor Report · eLife assessment]

This is an **important** study, that adds to the field a new understanding of exercise or mechanical loading, microRNAs, and secreted extracellular vessicles in the field of lung cancer (NSCLC), which may have relevance to other osteolytic cancers. The strength of the evidence was mixed: whereas in vitro microRNA experiments were **convincing**, other elements were **incomplete** (e.g., proving the roles of osteocytes, as opposed to other mechanosensitive cells, in vivo). This work would be of broad interest to those investigating osteolytic cancers, and the role of exercise in bone cancer, preclinically.

---

## [Referee Report · Reviewer #1 (Public Review)]

Xie and Colleagues propose here to investigate the mechanism by which exercise inhibits bone metastasis progression. The authors describe that osteocyte, sensing mechanical stimulation generated by exercise, inhibit NSCLC cell proliferation and sustain the dormancy thereof by releasing sEVs with tumor suppressor microRNAs. Furthermore, mechanical loading of the tibia inhibited the bone metastasis progression of NSCLC. Interestingly, exercise preconditioning effectively suppressed bone metastasis progression.

---

## [Referee Report · Reviewer #2 (Public Review)]

In this manuscript, Xie and colleagues investigate the contribution of osteocytes to bone metastasis of non-small cell lung carcinoma (NSCLC) using a combination of clinical samples and in vitro and in vivo data. They find that metastatic NSCLC cells exhibit lower levels of the proliferation markers Ki-67 and CCND3 when located in areas adjacent to the bone surface in both NSCLC patients and an intraosseous animal model of NSCLC. Using in vitro approaches, they show that osteocyte-like cells inhibit the proliferation of NSCLC cells through the secretion of small extracellular vesicles (sEVs). They identify miR-99b-3p as a component of sEVs and demonstrate that miR-99b3p inhibits the proliferation of NSCLC cells by targeting the transcription factor MDM2. Interestingly, the data also shows that mechanical stimulation of osteocytes enhances the inhibitory effect of osteocytes on NSCLC cell proliferation via increasing sEVs release. By performing different in vivo studies, the authors show that tibial loading and moderate exercise (treadmill running), before and after tumor cell inoculation, suppress tumor progression in bone and protect bone mass. Intriguingly, the moderate exercise regime shows additive/synergistic effects with the co-administration of anti-resorptive therapy. These data add to the growing evidence pointing towards osteocytes as important cells of the tumor microenvironment capable of influencing the progression of tumors in bone.

The conclusions of the paper, however, are not well supported by the data, and some critical aspects of image analysis and data analysis need to be clarified and extended.

(1) In Figure 1, the authors rely on KI-67 as a marker of proliferation. Yet, it is intriguing that some osteocytes, non-proliferating cells by definition, are often positive for this marker, which questions the specificity of the staining. The data displayed in supplementary figures showing CCND3 as a marker of proliferation ,and GFP as a marker of cancer cells, is much more robust and should be moved to the main figures.

(2) Adding control groups to fully assess the impact of the in vivo interventions (tibial loading, moderate exercise, anti-resorptive therapy) on bone mass would be needed. The authors should have used naive mice or analyzed the bones from the non-injected contralateral legs.

(3) The data on miRNA99b-3p on NSCLC in Supplementary figure 3 is not convincing. The positive cells are difficult to see and most of the osteocyte lack nuclei. Better data, in humans and the mouse model, would have helped to confirm that osteocytes produce miRNA99b-3p.

(4) Some conclusions of the paper are not entirely supported by the data provided. Osteocytes, as well as other bone cells, can respond to mechanical stimulation and thus could virtually be responsible for the protective effects of mechanical loading or moderate exercise. While blocking miR-99b3p with antagomiRs rescued the decreases in proliferation, it is unclear whether this effect is mediated by osteocytes or other cells that express this miRNA. In vivo experiments demonstrating a direct role of osteocytes are needed to support the notion that osteocytes maintain tumor dormancy in NSCLC bone metastasis. In vivo, studies assessing tumor dormancy directly would be needed to confirm osteocytes promote cancer cell dormancy.

---

## [Author Response]

The following is the authors’ response to the original reviews.

**Recommendations for the authors**

**Reviewer #1 (Recommendations For The Authors):**
(1) Please expand methods with additional details related to cell co-culture, such as cell numbers and duration.

We thank the reviewer for the careful reading and constructive suggestions and we are sorry to make you confused. We have added the experimental details (manuscript line 551-553) related to co-culture in the revised manuscript.

(2) Please unify the writing of the abbreviation of small extracellular vesicles in the text, figure, and caption.

Thank you for your comments. We have unified the abbreviation of extracellular vesicles to sEVs in the revised manuscript.

(3) The effects of components other than sEVs in mechanically stimulated osteocyte CM on the proliferation of NSCLC cells should be evaluated.

We evaluated the effects of SF, lEVs and sEVs in osteocyte CM on NSCLC cell proliferation under mechanical stimulation, and found that sEVs had the most obvious inhibition on NSCLC cell proliferation, as shown in the revised Supplemental Figure 4c, d.

(4) In addition to osteocytes and osteoblasts, the effects of other types of cells on the proliferation of NSCLC cells should be detected. It is recommended to add at least one type of cell from an infrequent metastatic site of NSCLC as a negative control.

We thank the reviewer for the suggestion. We added NCM460 cell line (derived from intestinal epithelium) as a negative control and found that NCM460 had no significant effect on NSCLC cell proliferation, as shown in Figure 1d. These experiments were conducted before our lastsubmission.

(5) The bone microenvironment is complex. It is recommended to evaluate the effect of bone marrow-derived sEVs on NSCLC to validate whether the tumor suppressive effect of osteocyte sEVs is unique.

We thank the reviewer for the suggestion. We agree with the reviewer’s comments that the bone microenvironment is complex. We explored the effect of bone marrow-derived sEVs on NSCLC cell proliferation and found that bone marrow-derived sEVs promoted NSCLC cell proliferation, as shown in Supplemental Figure 2g, h in the revised manuscript.

(6) The description of exercise preconditioning is not clear enough. It is recommended to supplement the pattern diagram to improve readability. Exercise preconditioning should be further discussed by the Authors.

Thank you for your comments and we are sorry to make you confused. We have added the pattern diagram of the exercise preconditioning in Supplemental Figure 6a.

**Reviewer #2 (Recommendations For The Authors):**
(1) The histological images are analyzed in a qualitative manner, with no description of the methodology used. A quantitative assessment of the distance and level of Ki-67+ NSCLC cells needs to be performed in human and murine tissues. Because in bone metastases cancer cells are frequently mixed with bone marrow cells, the inclusion of a cell marker to identify NSCLC cells is needed for proper interpretation of the imaging data.

We thank the reviewer for the careful reading and constructive suggestions. We conducted the suggested quantitative assessment and descripted the methodology in the revised manuscript. The results showed that Ki-67 was lower in tumor cells adjacent to bone tissue than in the surrounding tumor cells (Figure 1a, b).

In order to effectively identify NSCLC cells in bone metastases, GFP-expressing NSCLC cells were used in the animal model. We have added the immunofluorescence analysis of GFP and CCND3 in Supplemental Figure 4e, 4g, 5 and 6b.

(2) The authors rely on KI-67 as a marker of proliferation. Yet, it is intriguing that some osteocytes, non-proliferating cells by definition, are often positive for this marker, which questions the specificity of the staining. The authors should provide the proper immunostaining controls to check for specificity and use additional markers of proliferation to confirm these results.

We thank the reviewer for the suggestions. Ki-67 staining was wildly used to determine the dormancy of tumor cells in previous studies [1-4]. To confirm the results of Ki-67 staining, we used cyclin D3 (CCND3) as an additional marker of proliferation as suggested by the reviewer. We added the immunofluorescence analysis of CCND3 in Supplemental Figure 4e, 4g, 5 and 6b, which is consistent with the result of the quantitative immunofluorescence analysis of KI-67.

(3) The lack of proper controls in the in vivo experiments makes the interpretation of the data difficult. For instance, in the preconditioning experiment, it is likely that the bone mass increases. thus, these mice start with high bone mass than the control mice. The lack of a proper control (naive mice exposed to moderate exercise) does not allow testing if the presence of cancer cells still promotes bone loss in this group. The authors need to include naive mice or analyze the bones from the non-injected contralateral legs.

We thank the reviewer for the thoughtful comments and we are sorry to make you confused. We absolutely agree with the reviewer that the bone mass increases after exercise preconditioning.Multiple tissues and organ systems are affected by exercise, initiating diverse homeostatic responses. Although exercise preconditioning effectively suppressed bone metastasis progression of NSCLC as mentioned in the previous manuscript, we cannot immediately conclude that it is completely dependent on osteocytes to function. The mechanism of exercise preconditioning in suppressing bone metastasis progression is complex which still need further exploration. The revised manuscript has expanded the discussion on this area (manuscript line 326-328).

(4)Further, validating the in vivo work with other osteocyte-like cells or primary osteocytes would have strengthened the results.

We thank the reviewer for the suggestion. We have conducted the experiments of co-culture ofMLO-A5 (another type of osteogenic cell line) and NSCLC cells as shown in SupplementalFigure 1g. Not surprisingly, MLO-A5 cells also had an inhibitory effect on proliferation of NSCLC cells.

(5) The data on miRNA99b-3p on NSCLC in Supplementary Figure 3 is not convincing. The positive cells are difficult to see and most of the osteocyte lack nuclei. Better data, in humans and the mouse model, is needed to confirm that osteocytes produce miRNA99b-3p.

We thank the reviewer for the comments and we are sorry to make you confused. In this study, we used miRCURY LNA miRNA detection probes in ISH without staining the nuclei in the tissues, which method have been used in our previous studies with others [5-7]. Detailed experimental procedures for ISH of miRNA have been added in the revised manuscript (manuscript line461-474).

(6) The authors do not provide a piece of data supporting that osteocytes are responsible for any of the effects seen by the interventions done in the in vivo models. Osteocytes, as well as other bone cells, can respond to mechanical stimulation and thus could virtually be responsible for the protective effects of mechanical loading or moderate exercise. In vivo experiments demonstrating a direct role of osteocytes-produced miRNA99b-3p are needed to support the notion that osteocytes maintain tumor dormancy in NSCLC bone metastasis.

We thank the reviewer for the thoughtful comments and suggestion. We constructed in vivo model by injecting with antagomir-NC and antagomir-99b-3p with mechanical loading [8]. The results showed that the injection of antagomiR-99b-3p could partially and effectively rescue the inhibitory effect on NSCLC cell proliferation (Figure 4i-k).

(7) Further, the authors solely rely on Ki-67 as a marker of dormancy. Completing this analysis with an assessment of a dormant gene expression signature or in vivo studies assessing tumor dormancy directly would be needed to confirm this notion.

We thank the reviewer for the suggestion. We conducted the suggested experiment by using CCND3 as an additional dormancy marker. We added the immunofluorescence analysis of CCND3 in Supplemental Figure 4e, 4g, 5 and 6b, which is consistent with the result of the quantitative immunofluorescence analysis of Ki-67.

References

[1] Guba M, Cernaianu G, Koehl G et al. A primary tumor promotes dormancy of solitary tumor cells before inhibiting angiogenesis. Cancer Res, 2001, 61: 5575-9.

[2] Bliss Sarah A, Sinha Garima, Sandiford Oleta A et al. Mesenchymal Stem Cell-DerivedExosomes Stimulate Cycling Quiescence and Early Breast Cancer Dormancy in Bone Marrow. Cancer Res, 2016, 76: 5832-5844.

[3] Correia Ana Luísa, Guimaraes Joao C, Auf der Maur Priska et al. Hepatic stellate cells suppress NK cell-sustained breast cancer dormancy. Nature, 2021, 594: 566-571.

[4] Hu Jing, Sánchez-Rivera Francisco J, Wang Zhenghan et al. STING inhibits the reactivation of dormant metastasis in lung adenocarcinoma. Nature, 2023, 616: 806-813.

[5] Song Qiancheng, Xu Yuanfei, Yang Cuilan et al. miR-483-5p promotes invasion and metastasis of lung adenocarcinoma by targeting RhoGDI1 and ALCAM. Cancer Res, 2014, 74: 3031-42.

[6] Carotenuto Pietro, Hedayat Somaieh, Fassan Matteo et al. Modulation of Biliary CancerChemo-Resistance Through MicroRNA-Mediated Rewiring of the Expansion of CD133+ Cells. Hepatology, 2020, 72: 982-996.

[7] Lv Yan, Wang Yin, Song Yu et al. LncRNA PINK1-AS promotes Gαi1-driven gastric cancer tumorigenesis by sponging microRNA-200a. Oncogene, 2021, 40: 3826-3844.

[8] Zhang Yun, Li Shuaijun, Jin Peisheng et al. Dual functions of microRNA-17 in maintaining cartilage homeostasis and protection against osteoarthritis. Nat Commun, 2022, 13: 2447.